# Modulating transfer between tasks in gradient-based meta-learning

## Abstract

*Learning-to-learn* or *meta-learning* leverages data-driven inductive bias to increase the efficiency of learning on a novel task. This approach encounters difficulty when transfer is not mutually beneficial, for instance, when tasks are sufficiently dissimilar or change over time. Here, we use the connection between gradient-based meta-learning and hierarchical Bayes (Grant et al., 2018) to propose a mixture of hierarchical Bayesian models over the parameters of an arbitrary function approximator such as a neural network. Generalizing the model-agnostic meta-learning (MAML) algorithm (Finn et al., 2017), we present a stochastic expectation maximization procedure to jointly estimate parameter initializations for gradient descent as well as a latent assignment of tasks to initializations. This approach better captures the diversity of training tasks as opposed to consolidating inductive biases into a single set of hyperparameters. Our experiments demonstrate better generalization performance on the standard *mini*ImageNet benchmark for 1-shot classification. We further derive a novel and scalable non-parametric variant of our method that captures the evolution of a task distribution over time as demonstrated on a set of few-shot regression tasks.

## 1 Introduction

Meta-learning algorithms aim to increase the efficiency of learning by treating task-specific learning episodes as examples from which to generalize (Schmidhuber, 1987). The central assumption of a meta-learning algorithm is that some tasks are inherently related and so inductive transfer can improve generalization and sample efficiency (Caruana, 1993; 1998; Baxter, 2000). Recent meta-learning algorithms have encoded this assumption by learning global hyperparameters that provide a task-general inductive bias. In learning a single set of hyperparameters that parameterize, for example, a metric space (Vinyals et al., 2016) or an optimizer for gradient descent (Ravi & Larochelle, 2017; Finn et al., 2017), these meta-learning algorithms make the assumption that tasks are equally related and therefore mutual transfer is appropriate. This assumption has been cemented in recent few-shot learning benchmarks, which consist of a set of tasks generated in a systematic manner (*e.g.,* Finn et al., 2017; Vinyals et al., 2016).

However, the real world often presents scenarios in which an agent must decide what degree of transfer is appropriate. In the case of positive transfer, a subset of tasks may be more strongly related to each other and so non-uniform transfer poses a strategic advantage. Negative transfer in the presence of dissimilar or outlier tasks worsens generalization performance (Rosenstein et al., 2005). Moreover, when the underlying task distribution is non-stationary, inductive transfer to initial tasks should exhibit graceful degradation to address the catastrophic forgetting problem (Kirkpatrick et al., 2017). However, the consolidation of all inductive biases into a single set of hyperparameters cannot flexibly account for variability in the task distribution. In contrast, in order to deal with this degree of task heterogeneity, extensive task-switching literature reveals that people detect and readily adapt even in the face of significantly novel contexts (see Collins & Frank, 2013, for a review).

In this work, we learn a mixture of hierarchical models that allows the meta-learner to adaptively select over a set of learned parameter initializations for gradient-based fast adaptation (Finn et al., 2017) to a new task. The method is equivalent to clustering task-specific parameters in the hierarchical model induced by recasting gradient-based meta-learning as hierarchical Bayes (Grant et al., 2018) and generalizes the model-agnostic meta-learning (MAML) algorithm introduced in Finn et al. (2017).

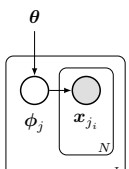 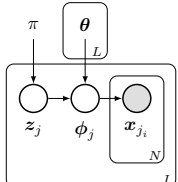 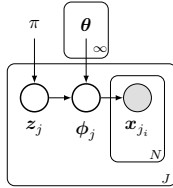

**Figure 1:** (a) The standard hierarchical Bayesian model for multi-task learning. A set of global parameters $\boldsymbol{\theta}$ provides an inductive bias for the estimation of task-specific parameters $\boldsymbol{\phi}_j$. (b) In a mixture of hierarchical Bayesian models, the cluster assigment of each task-specific parameter set $\boldsymbol{\phi}_j$ is represented with a latent Categorical variable $\boldsymbol{z}_j$. (c) Allowing an unbounded number of mixture components instantiates a non-parametric model that has the potential to grow with the data in complexity.

By treating the assignment of task-specific parameters to clusters as latent variables in a probabilistic model, we can directly detect similarities between tasks on the basis of the task-specific likelihood, which may be parameterized by a black-box model such as a neural network. Our approach therefore alleviates the need for explicit geometric or probabilistic modelling assumptions about the weights of a parametric model and provides a scalable method to regulate information transfer between episodes.

We extend our latent variable model to the non-parametric setting and leverage stochastic point estimation for scalable inference in a Dirichlet process mixture model (DPMM) (Rasmussen, 2000). To the best of our knowledge, no previous work has considered a scalable stochastic point estimation in a non-parametric mixture model. Furthermore, we are not aware of prior work applying non-parametric mixture modelling techniques to high-dimensional parameter spaces such as those of deep neural networks. The non-parametric extension allows the complexity of a meta-learner to evolve by introducing or removing clusters in alignment with the changing composition of the dataset and preserves performance on previously encountered tasks better than a parametric counterpart.

## 2 GRADIENT-BASED META-LEARNING AS EMPIRICAL BAYES

The goal of a meta-learner is to extract task-general knowledge from the experience of solving a number of related tasks. By leveraging this acquired prior knowledge, the meta-learner can quickly adapt to novel tasks even in the face of limited data or limited computation time (Schmidhuber, 1992). Recent approaches to meta-learning consolidate information from a set of training tasks into the parameters of a mapping to be applied at test time to a novel task. This mapping has taken the form of, for instance, a learned metric space (*e.g.,* Vinyals et al., 2016; Snell et al., 2017), a trained recurrent neural network (*e.g.,* Santoro et al., 2016), or an gradient-based optimization algorithm with learned parameters (*e.g.,* Ravi & Larochelle, 2017).

Model-agnostic meta-learning (MAML) (Finn et al., 2017) is a gradient-based meta-learning approach that estimates global parameters to be shared among task-specific models as an initialization for a few steps of gradient descent. MAML also admits a natural interpretation as parameter estimation in a hierarchical model, where the learned initialization acts as data-driven regularization for task-specific parameters (Grant et al., 2018). In particular, Grant et al. (2018) cast MAML as posterior inference for task-specific parameters $\boldsymbol{\phi}_j$ given a batch of task data $\boldsymbol{x}_{j_{1:N}}$ and a prior over $\boldsymbol{\phi}_j$ that is induced by early stopping of an iterative descent procedure. A few steps of gradient descent on the negative log-likelihood $-\log p(\boldsymbol{x}_{j_{1:N}} \mid \boldsymbol{\phi}_j^{(\ell)})$, starting from $\boldsymbol{\phi}_j = \boldsymbol{\theta}$ can be then understood as mode estimation of the posterior $p(\boldsymbol{\phi}_j^{(\ell)} \mid \boldsymbol{x}_{j_{1:N}}, \boldsymbol{\theta})$. The mode estimates $\hat{\boldsymbol{\phi}}_j = \boldsymbol{\theta} + \alpha \nabla_{\boldsymbol{\theta}} \log p(\boldsymbol{x}_{j_{1:N}} \mid \boldsymbol{\theta})$ are then combined to evaluate the marginal likelihood as $p(\{\boldsymbol{x}_{1:M}\}_{j=1}^J \mid \boldsymbol{\theta}) = \prod_j \int p(\boldsymbol{x}_{1:M} \mid \boldsymbol{\phi}_j) p(\boldsymbol{\phi}_j \mid \boldsymbol{\theta}) \, \mathrm{d}\boldsymbol{\phi}_j \approx \prod_j p(\boldsymbol{x}_{1:M} \mid \hat{\boldsymbol{\phi}}_j)$, where $\boldsymbol{x}_{j_{1:M}}$ is another batch of data from each task. A training dataset can then be summarized in an empirical Bayes point estimate of $\boldsymbol{\theta}$ computed by gradient descent in $-\log p(\{\boldsymbol{x}_{1:M}\}_{j=1}^J \mid \boldsymbol{\theta})$ (Lehmann & Casella, 2006). As such, the likelihood of a datapoint $\boldsymbol{x}$ sampled from a new task depends only on the empirical Bayes estimate of $\boldsymbol{\theta}$.

---

**Algorithm** ( $\mathscr{D}$, $\beta$, $\alpha$, $\tau$, $L_0$, $J$, $N$, $M$, $K$, $G_0$ )

 Initialize $L \leftarrow L_0$ and $\{\boldsymbol{\theta}^{(1)}, \ldots, \boldsymbol{\theta}^{(L)}\}$ as $\boldsymbol{\theta}^{(\ell)} \sim G_0$ for $\ell$ in $1, \ldots, L$

 **while** *not converged* **do**

  Draw tasks $\mathcal{T}_1, \ldots, \mathcal{T}_J \sim p_{\mathscr{D}}(\mathcal{T})$

  **for** $j$ *in* $1, \ldots, J$ **do**

   Draw task-specific datapoints $\boldsymbol{x}_{j_1}, \ldots, \boldsymbol{x}_{j_N}, \boldsymbol{x}_{j_{N+1}}, \ldots \boldsymbol{x}_{j_{N+M}} \sim p_{\mathcal{T}_j}(\boldsymbol{x})$

   $\left(\text{Draw a parameter initialization for a new cluster } \boldsymbol{\theta}^{(L+1)} \sim G_0\right)$

   **for** $\ell$ *in* $\{1, \ldots, L, (L+1)\}$ **do**

    Set $\boldsymbol{\phi}_j^{(\ell)} \leftarrow \boldsymbol{\theta}^{(\ell)}$ as the initialization for gradient-based fast adaptation

    Compute mode estimate via $\boldsymbol{\phi}_j^{(\ell)} \leftarrow \boldsymbol{\phi}_j^{(\ell)} + \alpha \nabla_{\boldsymbol{\phi}} \sum_i \log p(\boldsymbol{x}_{j_i} \mid \boldsymbol{\phi}_j^{(\ell)})$ for $K$ steps

   Compute assignment of tasks to clusters as $\gamma_j \leftarrow \texttt{E-STEP}\left(\{\boldsymbol{x}_{j_i}\}_{i=1}^N, \{\boldsymbol{\phi}_j^{(\ell)}\}_{\ell=1}^L\right)$

  Update $\boldsymbol{\theta}^{(\ell)} \leftarrow \boldsymbol{\theta}^{(\ell)} + \texttt{M-STEP}\left(\{\{\boldsymbol{x}_{j_{N+i}}\}_{i=1}^M, \boldsymbol{\phi}_j^{(\ell)}, \gamma_j\}_{j=1}^J\right)$ for $\ell$ in $1, \ldots, L$

  $\left(\text{Update global prior } G_0 \text{ with statistics based on } G_0 \text{ and } \{\boldsymbol{\theta}_1, \ldots\}\right)$

 **return** $\{\boldsymbol{\theta}^{(1)}, \ldots\}$

---

**Algorithm 2:** Stochastic expectation maximization via gradient-based meta-learning for clustering of task-specific parameters in the few-shot learning setting. (Commands in parentheses are executed only for the nonparametric variant of the algorithm.)

| $\texttt{E-STEP}\left(\{\boldsymbol{x}_{j_i}\}_{i=1}^N, \{\boldsymbol{\phi}_j^{(\ell)}\}_{\ell=1}^L\right)$ | $\texttt{M-STEP}\left(\{\boldsymbol{x}_{j_i}\}_{i=1}^M, \boldsymbol{\phi}_j^{(\ell)}, \gamma_j\right)$ |
|---|---|
|  **return** $\tau\text{-softmax}_\ell(\sum_i \log p(\boldsymbol{x}_{j_i} \mid \boldsymbol{\phi}_j^{(\ell)}))$ |  **return** $\beta \nabla_{\boldsymbol{\theta}}[\sum_{j,i} \gamma_j \log p(\boldsymbol{x}_{j_i} \mid \boldsymbol{\phi}_j^{(\ell)})]$ |

**Subroutine 3:** The E-STEP and M-STEP for a finite mixture of hierarchical Bayesian models.

## 3   LEARNING LATENT TASK STRUCTURE WITH GRADIENT-BASED META-LEARNING

If the task distribution is heterogeneous, assuming a single parameter initialization $\boldsymbol{\theta}$ is not suitable because it is unlikely that the point estimate computed by a few steps of gradient descent will sufficiently adapt the task-specific parameters $\boldsymbol{\phi}$ to a diversity of tasks. Moreover, explicitly estimating relatedness between tasks has the potential to aid the efficacy of a meta-learning algorithm by modulating both positive and negative transfer (Thrun; Zhang & Schneider, 2010; Rothman et al., 2010; Zhang & Yeung, 2014; Xue et al., 2007). Nonetheless, defining an appropriate notion of task relatedness is a difficult problem in the high-dimensional parameter or activation space of models such as neural networks.

Using the probabilistic interpretation of Section 2, we may deal with the variability in the tasks $\mathcal{T}_j$ by assuming that each set of task-specific parameters $\boldsymbol{\phi}_j$ is drawn from a mixture of base distributions each of which is parameterized by a hyperparameter $\boldsymbol{\theta}^{(\ell)}$. Accordingly, we capture task relatedness by estimating the likelihood of assigning each task to a mixture component based only on the task loss itself after a single step of fast adaptation (Finn et al., 2017). The result is a scalable end-to-end meta-learning algorithm that jointly learns task-specific cluster assignments and network parameters. This algorithm, further detailed in the following section, is capable of modulating the transfer of information across tasks to better generalize to heterogeneous or evolving task distributions.

### 3.1   EXPECTATION MAXIMIZATION FOR GRADIENT-BASED META-LEARNING WITH LATENTS

Let $\boldsymbol{z}_j$ be the Categorical latent variable indicating the cluster assignment of each task-specific parameter $\boldsymbol{\phi}_j$. A direct maximization of the mixture model likelihood is a combinatorial optimization problem that can grow intractable. This intractability is equally problematic for the posterior distribution over the cluster assignment variables $\boldsymbol{z}_j$ and the task-specific parameters $\boldsymbol{\phi}_j$, which are both treated as latent variables in the probabilistic formulation of meta-learning.

A standard approach for estimation in latent variable models such as probabilistic mixtures is to represent the distribution using samples with a sampler. The most widely used is the Gibbs

| Model | 1-shot | | |
|---|---|---|---|
| **matching network FCE** (Vinyals et al., 2016)[a] | 43.56 | ± | 0.84 |
| **meta-learner LSTM** (Ravi & Larochelle, 2017) | 43.44 | ± | 0.77 |
| **SNAIL** (Mishra et al., 2018)[b] | 45.1 | ± | —— |
| **prototypical networks** (Snell et al., 2017)[c] | 46.61 | ± | 0.78 |
| **MAML** (Finn et al., 2017) | 48.70 | ± | 1.84 |
| **LLAMA** (Grant et al., 2018) | 49.40 | ± | 1.83 |
| **KNN + GNN embedding** (Garcia & Bruna, 2017) | 49.44 | ± | 0.28 |
| **mAP-DLM** (Triantafillou et al., 2017) | 49.82 | ± | 0.78 |
| **fwCNN (Hebb)** (Munkhdalai & Trischler, 2018) | 50.21 | ± | 0.37 |
| **GNN** (Garcia & Bruna, 2017) | 50.33 | ± | 0.36 |
| **Our method** (clustering all layers of a neural network) | 50.80 | ± | 1.70 |

**Table 1:** Performance according to meta-test accuracy on the *mini*ImageNet 5-way, 1-shot classification benchmark from Vinyals et al. (2016). We report further comparisons in Appendix A. [a] Results reported by Ravi & Larochelle (2017). [b] We report test accuracy for a comparable architecture. [c] We report test accuracy for models matching train and test "shot" and "way".

sampler (Neal, 2000; Gershman & Blei, 2012), which draws from the conditional distribution of each latent variable given the others until convergence to the posterior distribution over all the latents. However, in the setting of latent variables defined over high-dimensional parameter spaces such as those of neural network models, a sampling approach such as Gibbs sampling is prohibitively expensive (Neal, 2012; Müller & Insua, 1998).

Instead of maintaining samples to represent the distribution over the latent variables, a scalable approximation involves representing the conditional distribution for each latent variable with either a *maximum a posteriori* (MAP) value or an expectation. In our meta-learning setting of a mixture of hierarchical Bayesian models, this suggests an augmented expectation maximization (EM) procedure (Dempster et al., 1977) alternating between an E-STEP that computes an expectation of the task-to-cluster assignments, which itself involves the computation of a MAP estimate for the task-specific parameters, and an M-STEP that computes a local maximum of the hyperparameters $\theta^{(1:L)}$.

To ensure scalability, we use the minibatch variant of stochastic optimization (Robbins & Monro, 1951) to compute both the E-STEP and M-STEPs; such approaches to EM are motivated by a view of the algorithm as optimizing a single free energy at both the E-STEP and the M-STEP (Neal & Hinton, 1998). In particular, for each task $j$ and cluster $\ell$, we follow the gradients to minimize the negative log-likelihood on the training data points using the cluster parameters $\theta^{(\ell)}$ as initialization. This allows us to obtain a modal point estimate of the task parameters $\hat{\phi}_j^{(\ell)}$.

The E-STEP in Subroutine 3 leverages the connection between gradient-based meta-learning and hierarchical Bayes (HB) (Grant et al., 2018) to employ the task-specific parameters to compute the posterior probability of cluster assignment. Accordingly, based on the likelihood of the same training data points under the model parameterized by $\hat{\phi}_j^{(\ell)}$, we compute the cluster assignment probabilities as

$$\gamma_j^{(\ell)} := p\big(z_j = \ell \mid x_j, \theta^{(1:L)}\big) \propto \int p(x_j \mid \phi_j^{(\ell)})\, p(\phi_j^{(\ell)} \mid \theta^{(\ell)})\, d\phi_j^{(\ell)} \approx p(x_j \mid \hat{\phi}_j^{(\ell)}) \,.$$

The cluster means $\theta^{(\ell)}$ are then updated by gradient descent on the validation loss in the M-STEP, given in Subroutine 3; this M-STEP is similar to the MAML algorithm in Finn et al. (2017). Note that, unlike other recent approaches to probabilistic clustering (*e.g.*, Bauer et al., 2017) we adhere to the episodic meta-learning setup for both training and testing since only the task support set $x_{j_{1:N}}$ is used to compute both the point estimate $\hat{\phi}_j^{(\ell)}$ and the cluster responsibilities $\gamma_j^{(\ell)}$. See Algorithm 2 for the full algorithm, whose high-level structure is shared with the non-parametric variant of our method.

## 3.2 STANDARD FEW-SHOT CLASSIFICATION WITH *mini*IMAGENET

Clustering task-specific parameters provides a way for a meta-learner to deal with task heterogeneity since each cluster can be associated with a subset of the tasks that would benefit most from inductive

---

E-STEP ( $\{\boldsymbol{x}_{j_i}\}_{i=1}^N, \{\boldsymbol{\phi}_j^{(\ell)}\}_{\ell=1}^{L+1}$, *concentration $\zeta$, spawn threshold $\epsilon$*)

$\quad$ DPMM log-likelihood, $\rho_j^{(\ell)} \leftarrow \sum_i \log p(\boldsymbol{x}_{j_i} \mid \boldsymbol{\phi}_j^{(\ell)}) + \log n^{(\ell)}$ for all $\ell$ in $1, \ldots, L$

$\quad$ DPMM log-likelihood for new component, $\rho_j^{(L+1)} \leftarrow \sum_i \log p(\boldsymbol{x}_{j_i} \mid \boldsymbol{\phi}_j^{(L+1)}) + \log \zeta$

$\quad \gamma_j \leftarrow \tau\text{-softmax}(\rho_j^{(1)}, \ldots, \rho_j^{(L+1)})$

$\quad$ **if** $\gamma_j^{(L+1)} > \epsilon$ **then**

$\quad\quad$ Expand the model by incrementing $L \leftarrow L + 1$

$\quad$ **else**

$\quad\quad$ Renormalize $\gamma_j \leftarrow \tau\text{-softmax}(\rho_j^{(1)}, \ldots, \rho_j^{(L)})$

$\quad$ **return** $\gamma_j$

---

M-STEP ( $\{\boldsymbol{x}_{j_i}\}_{i=1}^M, \boldsymbol{\phi}_j^{(\ell)}, \gamma_j$, *concentration $\zeta$*)

$\quad$ **return** $\beta \nabla_{\boldsymbol{\theta}}[\sum_{j,i} \gamma_j \log p(\boldsymbol{x}_{j_i} \mid \boldsymbol{\phi}_j^{(\ell)}) + L \log \zeta + \sum_\ell \log \Gamma(n^{(\ell)})]$

---

**Subroutine 4:** The E-STEP and M-STEP for an infinite mixture of hierarchical Bayesian models.

transfer. We apply Algorithm 2 with $L = 5$ components to the 1-shot 5-way classification few-shot classification benchmark *mini*ImageNet (Vinyals et al., 2016) using the same data split, architecture, and hyperparameter values as in Finn et al. (2017). We additionally use $\tau = 1$ for the softmax temperature and the same initialization as Finn et al. (2017) for our global prior $G_0$ (which reduces to a fixed initialization in the parametric case).

While we do not expect the standard *mini*ImageNet dataset to present much heterogeneity given the uniform sampling assumptions behind its design, we demonstrate in Table 1 that our parametric meta-learner can improve the generalization of gradient-based meta-learning on this task. This result suggests some level of cluster differentiation even on this non-heterogeneous benchmark.

## 4    SCALABLE NON-PARAMETRIC MIXTURES FOR EVOLVING TASKS

The mixture of meta-learners developed in Section 3 addresses a drawback of meta-learning approaches such as MAML that consolidate task-general information into a single set of hyperparameters, but adds another dimension to model validation in the form of identifying the correct number of mixture components. While this may be resolved by cross-validation if the dataset is static and therefore the number of components may be fixed, adhering to a fixed number of components throughout training is not appropriate in the nonstationary regime, where the underlying task distribution changes and therefore different types of tasks are presented sequentially.

In this regime, it is important to add mixture components sequentially to enable specialization of the component meta-learners to the different types of tasks that constitute the dataset. To address this, we derive a scalable stochastic estimation procedure to compute the expectation of task-to-cluster assignments (E-Step) for a growing number of task clusters using an *infinite* or *non-parametric* mixture model (Rasmussen, 2000) called the Dirichlet process mixture model (DPMM). This approach obviates the need for an *a priori* fixed number of components and enables the model to unboundedly adapt its complexity according to the observed data.

The formulation of the DPMM that is most appropriate for incremental learning is the sequential draws formulation that corresponds to an instantiation of the Chinese restaurant process (CRP) (Rasmussen, 2000). A CRP prior over $\boldsymbol{z}$ allows some probability to be assigned to a new mixture component while the task identities are inferred in a sequential manner, and has therefore been key to recent online and stochastic learning of the DPMM (Lin, 2013). A draw from a CRP proceeds as follows: For a sequence of tasks $(1, \ldots, J)$, the first task is assigned to the first cluster and the $j$th subsequent task is then assigned to the $\ell$th cluster drawn with probability

$$p(\boldsymbol{z}_j = \ell \mid \boldsymbol{z}_{1:j-1}, \zeta) = \begin{cases} \frac{n^{(\ell)}}{n+\zeta} & \text{for} \quad \ell \leq L \\ \frac{\zeta}{n+\zeta} & \text{for} \quad \ell = L+1, \end{cases}$$

where $L$ indicates the number of non-empty clusters, $n^{(\ell)}$ indicates the number of tasks already occupying a cluster $\ell$, and $\zeta$ is a fixed positive concentration parameter. The prior probability

associated with a new mixture component is therefore $p(\boldsymbol{z}_j = L + 1 \mid \boldsymbol{z}_{1:j-1}, \zeta)$, and the joint log-likelihood of the infinite mixture model can be written as

$$\log p\left(\boldsymbol{x}_j, \boldsymbol{\phi}_{1:J} \mid \boldsymbol{\theta}^{(1:L)}\right) = \sum_\ell \sum_j \gamma_j^{(\ell)} \log p(\boldsymbol{x}_j, \boldsymbol{\phi}_j^{(\ell)} \mid \boldsymbol{\theta}^{(\ell)}) + L \log \zeta + \sum_\ell \log \Gamma(n^{(\ell)}),$$

where $\gamma_j^{(\ell)}$ is the responsibility of cluster $\ell$ for task-specific parameters $j$. We refer the reader to Rasmussen (2000) for more details on the likelihood function associated with the DPMM.

In a similar spirit to Section 3, we develop a stochastic EM procedure for this estimation problem. However, while the computation of the mode estimate of the task-specific parameters $\boldsymbol{\phi}$ is mostly unchanged from finite variant, the estimation of the cluster assignment variables $\boldsymbol{z}$ in the E-STEP requires revisiting the Gibbs conditional distributions due to the potential addition of a new cluster at each step. For a DPMM, the conditional distributions for $\boldsymbol{z}$ are

$$p\left(\boldsymbol{z}_{J+1} = \ell \mid \mathcal{T}_j, \boldsymbol{z}_{1:J}\right) \propto \begin{cases} n^{(\ell)} \int P(\boldsymbol{x}_{J+1} \mid \theta) dG_l(\theta) & \text{for} \quad \ell \leq L \\ \zeta \int P(\boldsymbol{x}_{J+1} \mid \theta) dG_0(\theta) & \text{for} \quad \ell = L + 1, \end{cases}$$

with $G_0$ as the base measure over the components of the CRP, also known as the global prior. $G_\ell$ is the prior over each cluster's parameters, initialized with a draw from a Gaussian centered at $G_0$ with a fixed variance.

Using a mode estimate $\hat{\boldsymbol{\phi}}_j$ for task-specific $\boldsymbol{\phi}_j$, the distribution over task-to-cluster assignments $\boldsymbol{z}_j$ can be expressed as

$$\log p\left(\boldsymbol{z}_{j+1} = \ell \mid \mathcal{T}_{j+1}, \boldsymbol{z}_{1:j}\right) \approx \begin{cases} \log n^{(\ell)} + \log p(\boldsymbol{x}_i \mid \hat{\boldsymbol{\phi}}) + \log p(\hat{\boldsymbol{\phi}} \mid \boldsymbol{\theta}^{(\ell)}) & \text{for} \quad \ell \leq L \\ \log \zeta + \log p(\boldsymbol{x}_i \mid \hat{\boldsymbol{\phi}}) + \log p(\hat{\boldsymbol{\phi}} \mid \boldsymbol{\theta}^{(0)}) & \text{for} \quad \ell = L + 1, \end{cases}$$

We thus obtain a set of local objective functions amenable to gradient-based optimization in a similar fashion to the parametric M-STEP of Subroutine 3. We can also omit the prior term $\log p(\hat{\boldsymbol{\phi}} \mid \boldsymbol{\theta}^{(\ell)})$ as it arises as an implicit prior resulting from truncated gradient descent, as explained in Section 3 of Grant et al. (2018).

One marked difference between the objective of the M-STEP in Subroutine 3 and that in Subroutine 4 is the penalty term of $\log n^{(\ell)}$ or $\log \zeta$ which incentivizes larger clusters in order to deter over-spawning. Accordingly, this approximate inference routine still preserves the preferential attachment ("rich get richer") dynamics of Bayesian nonparametrics (Raykov et al., 2016). Another is that is not immediate from the Gibbs conditionals is the use of a threshold on the cluster responsibilities to account for mini-batch noise when spawning a cluster based on a single batch. This threshold is necessary for our stochastic mode estimation of Algorithm 4 as it maintains that a new cluster's responsibility needs to exceed a certain value before being permanently added to the set of components (see the E-STEP in Subroutine 4 for more details). Intuitively, if a cluster has close to an equal share of responsibilities to existing clusters after accounting for the CRP penalty ($\log n^{(\ell)}$ vs. $\log \zeta$), it is spawned. A sequential approximation for nonparametric mixtures with a similar threshold was proposed in (Lin, 2013), where variational Bayes was used instead of point estimation in a DPMM.

Overall, unlike traditional nonparametric algorithms, our model does not refine the cluster assignments of previously observed points by way of multiple expensive passes over the whole data set. Instead, we incrementally infer model parameters and add components during episodic training based on noisy estimates but unbiased of the log-likelihood gradients. To the best of our knowledge, no previous work has considered a scalable stochastic point estimation of the parameters in a non-parametric mixture model. Furthermore, we are not aware of prior work applying non-parametric mixture models to high-dimensional function approximators such as modern neural networks with end-to-end learning.

## 5 "TASK-AGNOSTIC" FEW-SHOT LEARNING WITH AN EVOLVING DATASET

A non-parametric mixture model should be able to detect and adapt to a changing distribution of tasks, without any external information to signal the start of a new task type (*i.e.,* , in a "task-agnostic" manner), by adjusting its capacity. On the other hand, current meta-learning methods coerce

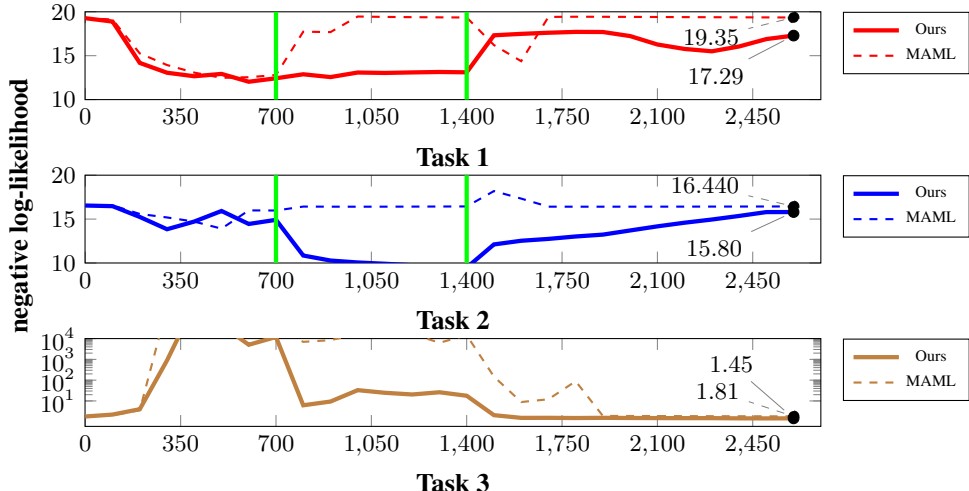

**Figure 5:** Mixture component validation log-likelihoods (MSE losses) on an evolving dataset that generates even polynomial regression tasks for 700 iterations, then odd polynomial regression tasks until iteration 1400 at which point it generates sinusoidal regression tasks. We plot the validation negative log likelihood of the data for each task. Note, for example, the change in loss to the second (red) cluster at 700 iterations when the odd polynomial tasks are introduced.

the inductive biases into a single set of hyperparameters and therefore inevitably saturate model parameters in the evolving dataset regime. Here, we present our main experimental results using synthetic few-shot regression tasks to demonstrate that our non-parametric meta-learning algorithm is able to acquire new inductive biases as new tasks are introduced without necessarily over-writing existing ones by adding new components.

## 5.1 SYNTHETIC REGRESSION TASKS: IMPROVED GENERALIZATION

**Experimental Setup**    To demonstrate the unconstrained and adaptive capacity of our non-parametric meta-learning model, we consider alternating sinusoidal, even-degree polynomial, and odd-degree polynomial regression tasks with input $x$ sampled uniformly from $[-5, 5]$ during the meta-training procedure. For the sinusoidal regression, we consider a sine wave with phase sampled uniformly from $[0, \pi]$ and with amplitudes sampled from $a_1 \sim \mathcal{N}(2, 1)$ in a similar fashion to the synthetic regression tasks in Finn et al. (2017)As for the polynomial regression, $y = \sum_i^d c_i x^i$ where $c_i \sim \mathcal{U}(-5, 5)$. For the experiment in Figure 5, we presented even-degree polynomial regression tasks for 700 iterations, followed by odd-degree polynomial regression tasks until 1400 iterations, before switching to sinusoidal regression tasks. We use the mean-squared error loss function for each task as the inner loop and meta-level objectives.

**Hyperparameter choices**    Our architecture is a feedforward neural network with 2 hidden layers with ReLU nonlinearities, each of size 40. We use a meta-batch size of 25 tasks (both for the inner updates and the meta-gradient updates) as in the setup for 10-shot sinusoidal regression in Finn et al. (2017). Our non-parametric algorithm starts with a single cluster ($L_0 = 1$ in Algorithm 4). In these experiments, we set the threshold $\epsilon = 0.95T/(L+1)$, with $L$ the number of non-empty clusters and $T$ the size of the meta-batch.

We also compute the cluster sizes using a moving window of size 20 (which is a dataset-dependent hyperparameter) to accommodate the order of magnitude in comparison to the small training losses used for cluster responsibility evaluation. This is necessary since we do not keep track of the exact prior cluster assignments for our randomly selected task mini-batches nor do we re-assess our assignments at each iteration. Otherwise, non-empty clusters can accumulate an extremely large number of assignments, making cluster spawning impossible after only a few meta-learning episodes. An additional practical reason for the stochastic setting is that it would be extremely expensive to store the assignments for an entire dataset in memory. Finally, preserving task assignments is potentially

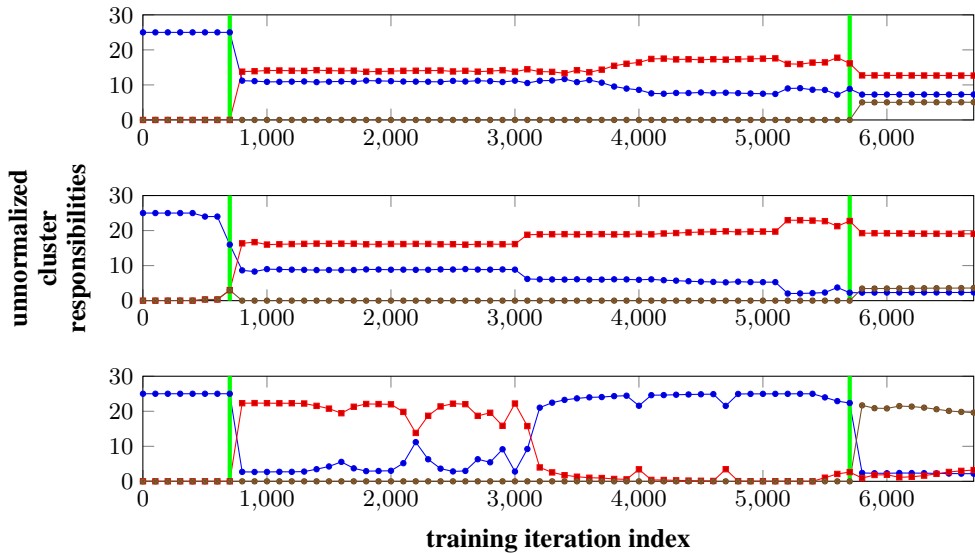

**Figure 6:** (Top) An evolving dataset of quadratic, sinusoidal, and logistic regression tasks. The dataset generates quadratic regression tasks for 700 iterations, then switches to generating sinusoidal regression tasks until iteration 5700 at which point it generates logistic regression tasks. (Below) Note the change in responsibilities to the second (red) cluster at 700 iterations when the sinusoidal tasks are introduced. At 5700 iterations, the logistic regression tasks (third row) also cause a third (brown) cluster to be introduced.

harmful due to stale parameter values since the task assignments in our framework are meant to be easily reconstructed on-the-fly using the `E-STEP` with updated parameters $\boldsymbol{\theta}$.

**Results.** In Figure 5, we report the mean-squared error (MSE) validation loss for each task as the 3 different tasks are introduced sequentially and disjoint (only one task for each training phase) to the non-parametric meta-learner. Overall, our algorithm consistently outperforms MAML in terms of validation loss on the three tasks across the three training phases. More interestingly, our algorithm preserves its performance on old tasks when switching to a new training phase, whereas MAML suffers from a clear degradation. While it seems our algorithm does not perfectly preserve old inductive biases, we can conclude from a parallel experiment in Figure 6 that it can increase its capacity, when needed, to adjust for new training phases. This allows for better preservation of previously learnt knowledge which is key for continual learning.

## 5.2 SYNTHETIC FEW-SHOT REGRESSION TASKS: TASK DIFFERENTIATION

In this section, we turn our focus to task differentiation. We thus investigate the cluster responsibilities on validation data from each of the 3 tasks: quadratic regression, sinusoidal regression, and logistic regression on data from the same input range as specified in Section 5.1.

In Figure 5, we notice a clear differentiation between the tasks as indicated by the cluster responsibilities. The responsibilities under the first cluster (in blue) decreases to almost zero for the 3rd task (sinusoid) while staying evenly split for the related odd and even polynomial regression tasks. Furthermore, a second cluster (in red) is spawned to account for the difference between odd and even degree polynomials. However, we also notice that the second cluster responsibilities are not zero for the first task, indicating similarities between even and odd polynomial regressions. The same behavior can be seen for the third cluster on the third task. Note that the sinusoidal regression task is the more difficult task which explains the different order of magnitude of the losses and the motivation for a longer period of training. Also note that regression losses are unbounded and thus pose difficulties to any optimization-based continual learner; accordingly, most continual learning datasets such as Moving MNIST consist of classification tasks that make use of a bounded cross-entropy error.

## 6 RELATED WORK

**Multi-task learning.** Rosenstein et al. (Rosenstein et al., 2005) demonstrated that negative transfer can worsen generalization performance. This has motivated much work on HB in transfer learning and domain adaptation (*e.g.*, Lawrence & Platt, 2004; Yu et al., 2005; Gao et al., 2008; Daumé III, 2009; Wan et al., 2012). Closest to our proposed approach is early work on hierarchical Bayesian multi-task learning with neural networks (Heskes, 1998; Bakker & Heskes, 2003; Salakhutdinov et al., 2013; Srivastava & Salakhutdinov, 2013). These approaches are different from ours in that they place a prior, which could be nonparametric as in Salakhutdinov et al. (2013) and Srivastava & Salakhutdinov (2013), only on the output layer. Furthemore, none of these approaches were applied to the episodic training setting of meta-learning. Heskes (1998) and Srivastava & Salakhutdinov (2013) also propose training a mixture model over the output layer weights using MAP inference. However, this approach does not scale well to all the layers in a network and performing full passes on the dataset for inference of the full set of weights can become computationally intractable.

**Continual learning.** Techniques developed specifically to address the catastrophic forgetting problem in continual learning, such as elastic weight consolidation (EWC) (Kirkpatrick et al., 2017), synaptic intelligence (SI) (Zenke et al., 2017), and variational continual learning (VCL) (Nguyen et al., 2017) require access to an explicit delineation between tasks that acts as a catalyst to grow model size, which we refer to as "task-aware." In contrast, our nonparametric algorithm tackles the "task-agnostic" setting of continual learning, where the meta-learner does not receive information about task changes but instead learns to recognize a shift in the task distribution and adapt accordingly.

**Clustering.** Incremental or stochastic clustering has been considered in the EM setting in Neal & Hinton (1998), and similarly for minibatch K-means (Sculley, 2010). Online learning of non-parametric mixture models is also a way of perform clustering in the mini-batch setting using sequential variational inference Lin (2013).A key distinction between our work and these approaches is that we leverage the connection between empirical Bayes in this model and gradient-based meta-learning (Grant et al., 2018) to use the MAML (Finn et al., 2017) objective as a log posterior surrogate. This allows our algorithm to scale and easily integrate with minibatch stochastic gradient-based meta-learning instead of alternating multiple backpropagation steps with multiple inference passes over the full dataset (Srivastava & Salakhutdinov, 2013; Bauer et al., 2017). Our approach is distinct from recent work on gradient-based clustering (Greff et al., 2017) since we adhere to the more challenging setting of episodic meta-learning for both training and testing. This can be a challenging setting for a clustering algorithm, as the assignments need to be computed using $K = 1$ examples per class as is the case in 1-shot learning.

## 7 CONCLUSION

Meta-learning is a source of learned inductive bias. Occasionally, the inductive bias is harmful because the experience gained from solving one task does not transfer well to another. On the other hand, if tasks are closely related, they can benefit from a greater amount of inductive transfer. Here, we present an approach that allows a gradient-based meta-learner to explicitly modulate the amount of transfer between tasks, as well as to adapt its parameter dimensionality when the underlying task distribution evolves. We formulate this as probabilistic inference in a mixture model that defines a clustering of task-specific parameters. To ensure scalability, we make use of the recent connection between gradient-based meta-learning and hierarchical Bayes (Grant et al., 2018) to perform approximate *maximum a posteriori* (MAP) inference in both a finite and an infinite mixture model. This approach admits non-conjugate likelihoods parameterised with a black-box function approximator such as a deep neural network, and therefore learns to identify underlying genres of tasks using the standard gradient descent learning rule. We demonstrate that this approach allows the model complexity to grow along with the evolving complexity of the observed tasks in both a few-shot regression and a few-shot classification problem.

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

## A    EXTENDED *mini*IMAGENET BENCHMARKING

| Model | 5-way acc. (%) 1-shot 5-shot | | | | | |
|---|---|---|---|---|---|---|
| fine-tuning [a] | 28.86 | ± | 0.54 | 49.79 | ± | 0.79 |
| nearest neighbor [a] | 41.08 | ± | 0.70 | 51.04 | ± | 0.65 |
| matching network FCE (Vinyals et al., 2016) [a] | 43.56 | ± | 0.84 | 55.31 | ± | 0.73 |
| meta-learner LSTM (Ravi & Larochelle, 2017) | 43.44 | ± | 0.77 | 60.60 | ± | 0.71 |
| SNAIL (Mishra et al., 2018) [b] | 45.1 | ± | —— | 55.2 | ± | —— |
| prototypical networks (Snell et al., 2017) [c] | 46.61 | ± | 0.78 | 65.77 | ± | 0.70 |
| MAML (Finn et al., 2017) | 48.70 | ± | 1.84 | 63.11 | ± | 0.92 |
| LLAMA (Grant et al., 2018) | 49.40 | ± | 1.83 | —— | ± | —— |
| mAP-DLM (Triantafillou et al., 2017) | 49.82 | ± | 0.78 | 63.70 | ± | 0.67 |
| KNN + GNN embedding (Garcia & Bruna, 2017) | 49.44 | ± | 0.28 | 64.02 | ± | 0.51 |
| GNN (Garcia & Bruna, 2017) | 50.33 | ± | 0.36 | 66.41 | ± | 0.63 |
| fwCNN (Hebb) (Munkhdalai & Trischler, 2018) | 50.21 | ± | 0.37 | 64.75 | ± | 0.49 |
| fwResNet (Hebb) (Munkhdalai & Trischler, 2018) | 56.84 | ± | 0.52 | 71.00 | ± | 0.34 |
| SNAIL (Mishra et al., 2018) | 55.71 | ± | 0.99 | 68.88 | ± | 0.92 |
| Gidaris & Komodakis | 56.20 | ± | 0.86 | 73.00 | ± | 0.64 |
| Bauer et al. (2017) | 56.30 | ± | 0.40 | 73.90 | ± | 0.30 |
| MetaNet (Munkhdalai & Yu, 2017) | 57.10 | ± | 0.70 | 70.04 | ± | 0.63 |
| MAML (Finn et al., 2017) [d] [e] | 58.05 | ± | 0.10 | 72.41 | ± | 0.20 |
| TADAM (Oreshkin et al., 2018) | 58.50 | ± | 0.30 | 76.70 | ± | 0.30 |
| Qiao et al. | 59.60 | ± | 0.41 | 73.74 | ± | 0.19 |
| LEO (Rusu et al., 2018) | 60.06 | ± | 0.05 | 75.72 | ± | 0.18 |
| Our method (clustering all layers) | 50.8 | ± | —- | —— | ± | —- |

**Table 2:** One-shot classification accuracy on the *mini*ImageNet test set, with comparison methods ordered by one-shot performance. All results are averaged over 600 test episodes, and we report $95\%$ confidence intervals when available. [a] Results reported by Ravi & Larochelle (2017). [b] We report test accuracy for a comparable architecture. [c] We report test accuracy for models matching train and test "shot" and "way". [d]We report test accuracy for a non-standard architecture with more parameters. [e]Results reported by Rusu et al. (2018).

figures/polye-polyo-sine-plot.tex

**Figure 7:** An evolving dataset of *mini*ImageNet few-shot classification tasks where for the first 20k iterations we train on the standard dataset, then switch to a "pencil" effect set of tasks for 10k iterations before finally switching to a "blurred" effect set of tasks until 40k. Responsibilities $\gamma^{(\ell)}$ for each cluster are plotted over time. Note the change in responsibilities as the dataset changes at iterations 20k and 30k.

## B  EVOLVING FEW-SHOT CLASSIFICATION

We apply Algorithm 4 to an evolving variant of the *mini*ImageNet few-shot classification dataset while using the same standard architecture of Vinyals et al. (2016); Finn et al. (2017). In this variant, different artistic filters are applied to the images in the few-shot classification tasks over the meta-training procedure to simulate a changing distribution of classification tasks. More specifically, we first train on the standard mini-imagenet tasks for 20000 iterations then introduce "pencil" effect tasks for 10000 iterations, before finally switching to a "radial blur" effect for another 10000 iterations.

Cluster responsibilities during training can be found in Figure 7. To compare more closely to the single-cluster baseline, we restrict the number of clusters to 1 for the first phase of training (the standard mini-imagenet tasks). However, the moment we start introducing new datasets, this restriction is lifted. This allows a better evaluation of the transfer from the first cluster to the new tasks as more datasets are introduced in an online setting.

We notice some differentiation between the tasks that is not as pronounced as what we observed on the toy data. This is potentially due to the fact that all the tasks are derived from the same core set of images. Accordingly, inductive biases learned on the unfiltered dataset for the first 20000 iterations can transfer to the filtered datasets more easily than the different regression tasks we experimented with.

