# OpenReview forum: "Modulating transfer between tasks in gradient-based meta-learning"
_ICLR.cc/2019/Conference_

### Official Review · AnonReviewer1 · 2018-10-25
**Gradient-base few-shot learning. Extends MAML to a mixture distribution, to allow for internal task clustering. Falls short of recent state-of-art results, while being even a lot slower than MAML**

**Rating:** 4
**Confidence:** 4

**Review:**

Summary:

This work tackles few-shot (or meta) learning, providing an extension of the gradient-based MAML method to using a mixture over global hyperparameters. Each task stochastically picks a mixture component, giving rise to task clustering. Stochastic EM is used for end-to-end learning, an algorithm that is L times more expensive than MAML, where L is the number of mixture components. There is also a nonparametric version, based on Dirichlet process mixtures, but a large number of approximations render this somewhat heuristic.

Comparative results are presented on miniImageNet (5-way, 1-shot). These results are not near the state-of-the art anymore, and some of the state-of-art methods are simpler and faster than even MAML. If expensive gradient-based meta-learning methods are to be consider in the future, the authors have to provide compelling arguments why the additional computations pay off.

- Quality: Paper is technically complex, but based on simple ideas. In the case of
   infinite mixtures, it is not clear what is done in the end in the experiments.
   Experimental results are rather poor, given state-of-the-art.
- Clarity: The paper is not hard to understand. What is done, is done cleanly.
- Originality: The idea of putting a mixture model on the global parameters is not
   surprising. Important questions, such as how to make this faster, are not
   addressed.
- Significance: The only comparative results on miniImageNet are worse than the
   state-of-the-art by quite a margin (admittedly, the field moves fast here, but it
   is also likely these benchmarks are not all that hard). This is even though better
   performing methods, like Versa, are much cheaper to run

While the idea of task clustering is potentially useful, and may be important in practical use cases, I feel the proposed method is simply just too expensive to run in order to justify mild gains. The experiments do not show benefits of the idea.

State of the art results on miniImageNet 5-way, 1-shot, the only experiments here which compare to others, show accuracies better than 53:
- Versa: https://arxiv.org/abs/1805.09921.
   Importantly, this method uses a simpler model (logistic regression head models)
   and is quite a bit faster than MAML, so much faster than what is proposed here
- BMAML: https://arxiv.org/abs/1806.03836.
   This is also quite complex and expensive, compared to Versa, but provides good
   results.

Other points:
- You use a set of size N+M per task update. In your 5-way, 1-shot experiments,
   what is N and M? I'd guess N=5 (1 shot per class), but what is M? If N+M > 5,
   then I wonder why results are branded as 5-way, 1-shot, which to mean means
   that each update can use exactly 5 labeled points.
   Please just be exact in the main paper about what you do, and what main
   competitors do, in particular about the number of points to use in each task
   update.
- Nonparametric extension via Dirichlet process mixture. This is quite elaborate, and
   uses further approximations (ICM, instead of Gibbs sampling).
   Can be seen as a heuristic to evolve the number of components.
   What is given in Algorithm 2, is not compatible with Section 4. How do you merge
   your Section 4 algorithm with stochastic EM? In Algorithm 2, how do you avoid
   that there is always one more (L -> L+1) components? Some threshold must be
   applied somewhere.
   An alternative would be to use split&merge heuristics for EM.
- Results reported in Section 5 are potentially interesting, but entirely lack a
   reference point. The first is artificial, and surely does not need an algorithm of this
   complexity. The setup in Section 5.2 is potentially interesting, but needs more
   work, in particular a proper comparison to related work.
   This type of effort is needed to motivate an extension of MAML which makes
   everything quite a bit more expensive, and lacks behind the state-of-art, which
   uses amortized inference networks (Versa, neural processes) rather than
   gradient-based.

---

> ### Author Response · Authors · 2018-11-22
> **[1/4] Thank you for your detailed feedback! Please let us know if there is more we can do to address your concerns.**
>
> We thank the reviewer for an extensive and detailed review. We respond below to some specific comments but please also refer to the general "response to all reviewers" above. We encourage the reviewer to follow up with any other points that would improve the paper.
>
>
> > "Stochastic EM is used for end-to-end learning, an algorithm that is L times more expensive than MAML, where L is the number of mixture components."
> > “Important questions, such as how to make this faster, are not addressed”
> > "[VERSA] uses a simpler model (logistic regression head models) and is quite a bit faster than MAML, so much faster than what is proposed here… [BMAML] is also quite complex and expensive, compared to Versa, but provides good results."
>
> We will clarify in a revised version of the paper that this approach is easily parallelizable by assigning the computation of the MAP estimate \hat{\phi}, as well as the computation of the gradient with respect to the hyperparameter \theta, to independent workers. Moreover, keeping the structure of the underlying probabilistic model fixed, our maximum a posteriori (MAP) procedure (which directly optimizes the negative log posterior via gradient descent) is the most straightforward approach to point estimation.
>
> In contrast, approaches like VERSA [Gor18] that use a hyper network to compute task-specific parameters or approaches that make use of amortized inference require a heavily parameterized hyper/inference network in order to compute the task-specific parameter values. The training of this hyper/inference network imposes an additional computational cost, even though test-time computation/inference of task-specific parameters can be performed via a single feedforward pas. As such, these different approaches present alternative trade-offs in speed at training versus test time.
>
> One modelling change we did experiment with on the homogeneous miniImageNet benchmark was learning the mixture model only on the last layer of the neural network initialization. In this case, we achieved significant speedups: The runtime for L = 2, …, 5 components was not much more than the original MAML runtime; this variant of our method is therefore quite simple and fast. The corresponding drop in generalization performance from clustering all the layers was not substantial. However, in the submission, we focused on the extensible (non)parametric mixture modeling aspect for the full set of neural network weights to demonstrate the generality and scalability of our method. Therefore, we did not prioritize reporting such results within the space constraints but will add them in an updated version of the paper.
>
> We welcome further clarification on drawbacks related to the runtime of our method with respect to alternative approaches. Are there further details we can provide?

---

> > ### Comment · AnonReviewer1 · 2018-11-26
> > **Reaction to author comments**
> >
> > I have read the (verbose) feedback of the authors, and hereby acknowledge that.
> >
> > My vote does not change. This paper provides a rather straightforward mixturization of MAML. This procedure makes the method substantially more expensive to run. On benchmarks the authors compared (very few), the payoff is minor, and the results are far from the state of art. The state of the art, such as VERSA etc, is also faster to run than even MAML, not to speak of the extension provided here. Results being equal, simplicity and ease of use of a method is the decisive factor.
> >
> > The authors claim their goal is not to outperform state of the art (or get close to it). But then, it is not clear to me what their goal is in the first place. I do not see why putting a mixture on top of MAML has a lot of potential for future improvements. The authors are free to convince the readers otherwise, by coming up with a relevant use case and demonstrate it. Their toy examples are not convincing. Putting so much complexity together should really be motivated by a convincing use case.
> >
> > Mixturizing probabilistic models is a pretty common step, I'd call it incremental. And where you like it or not, the Deep NN field (key audience for ICLR) is very much focused on competitive results.

---

> > > ### Author Response · Authors · 2018-12-05
> > > **Thank you for your response. [1/2]**
> > >
> > >  We thank the reviewer to the response. We respond to specific points below.
> > >
> > > > A “straightforward mixturization” is an understatement of the contribution of this work which does not recognize the difficulties of gradient-based clustering in both the stochastic, fast-adaptation, few-shot setting. We highlight some central challenges:
> > >
> > > _Stochastic setting_: Clustering in the stochastic setting with gradient information is not as straightforward as metric- or likelihood-based clustering that iteratively passes over the whole dataset. This is particularly a concern in the non-parametric setting which normally requires access to the set of all prior assignments (atoms) for the *entire* dataset in order to refine all assignments at each iteration (M-step). Even in the parametric setting, there is little work on *stochastic* gradient-based clustering (we can think of Neural EM [Gre17], which we cite in our paper).
> > >
> > > _Joint learning_: The fact that the reviewer refers to our mixture as “on top of MAML” indicates that our presentation might have not been clear enough: Our mixture learning approach is adaptive and takes place within gradient-based fast adaptation, not as an external step or loop over individual MAML components. In particular, the weighting of mixture components (E-step) takes into account the per-task loss and therefore allows appropriate information transfer between clusters for the meta-gradient step (M-step). We emphasize that this is not a simple categorization of the tasks after transfer has taken place, but an online procedure with dependencies.
> > >
> > > On the other hand, a "straightforward mixturization" could cluster the meta-parameters (\theta, not the task-specific weights \phi) after each component is trained independently. This does not address the issue of heterogeneity as it does not limit negative transfer or amplify positive transfer during training.
> > > Therefore, we can see the unique capacity of our EM-learning procedure is to cluster related tasks together to inform fast weight adaptation in a scalable manner.
> > >
> > > _No geometric assumptions/clustering based on optimal transfer instead of enforcing some notion of task similarity_: A "straightforward mixturization" could rely on the L2 distance among the cluster components (after both inner and update steps are taken). Not only would this approach encounter some scaling issues in the high-dimensional setting, but the L2 distance also does not account for the permutation of weights in a neural network, and could therefore erroneously differentiate functionally similar networks. This is also a limitation of likelihood-based clustering with standard (e.g., multivariate Normal) distributional assumptions. In contrast, our approach leverages the same computation used for gradient-based fast-adaptation to cluster based on the improved training loss, and to determine which tasks most benefit from mutual transfer.
> > >
> > > We do not view our approach as a "straightforward mixturization of MAML", nor do we think that "mixturizing" probabilistic models is "incremental". We view our submission as continuing a line of work that began with the identification of gradient-based meta-learning as parameter estimation in a hierarchical probabilistic model. We detail how we view this investigation in the author response entitled "Response to all reviewers: Alternate meta-learning algorithms correspond to the use of different inference techniques".
> > >
> > > > Notes on higher capacity / increased complexity:
> > > A higher capacity is unavoidable to address parameter saturation in an evolving setting. This is the main motivation for our approach. No prior work has addressed ways to optimally add or tune the capacity of a meta-learner, either in the stationary or non-stationary setting. Our approach allows adaptation of model capacity as needed based on the observed data. This cannot be achieved by cross-validation in the more difficult online setting.

---

> > > ### Author Response · Authors · 2018-12-05
> > > **Thank you for your response. [2/2]**
> > >
> > > > Comparison to VERSA:
> > > We were not aware of the per-iteration speed comparison since such results were only added recently (10 days before the revision deadline of Nov. 23) to the concurrent ICLR submission of VERSA ["Versatility" in Section 5.2, pg. 8]. However, seeing that the comparison merely adapted the public GitHub version of MAML, we would like to note that:
> > > - It is difficult to disentangle algorithmic and implementational speedups. Our own implementation of MAML is significantly faster than the standard version from the original author's GitHub repo, which is not heavily parallelized.
> > > - It is unclear if VERSA takes the same number of iterations during training (and thus the 5x speedup might not translate to 5x overall convergence speedup). This is related to a point that we make in the first author response: Approaches using amortization or hyper-networks trade off an increase in overall training time (due to extra parameters in the inference / hyper-network) for fast test-time inference. To fully disentangle these two components would be an orthogonal contribution. It is not the focus of our paper, as we focus on making the novel contribution of task-agnostic, continual meta-learning.
> > > - Our task-specific weights are a high-dimensional parameter set, unlike VERSA’s restriction to the last layer.
> > > - VERSA is *not* a comparable architecture to ours, as it uses 5 convolutional layers [see Table D.2, p. 19 of Gor18], as compared to 4 for the standard architecture, in addition to the extra parameters of the hyper-network.
> > > - Our model achieves SOTA with the standard architecture of 4 convolutional layers [Vin16]. Amongst all reported approaches with various architectures, VERSA’s 53.4% 5-way 1-shot accuracy is nowhere near the state-of-the-art, which uses residual networks [Gid18, Mun18a, Mun18b, Ore18, Rus18].
> > >
> > > > “it is not clear to me what their goal is in the first place”
> > > We believe we made it clear in the submission and rebuttal that our main goal is addressing the continual setting of meta-learning that focuses on the question of learning-to-learn in a non-stationary environment. Towards that goal, our regression tasks are in line with prior work and fully demonstrate:
> > > - a considerable amount of task differentiation (from the assignments plot);
> > > - adaptive model capacity (clusters were indeed spawned); and
> > > - improved generalization and less catastrophic forgetting (improved loss values for older tasks while training on newer tasks as compared to MAML).
> > >
> > > Lastly, we would like to point the reviewer to the ICLR 2019 reviewer guidelines (https://iclr.cc/Conferences/2019/Reviewer_Guidelines) in response to the following statement of R1: "And where [sic] you like it or not, the Deep NN field (key audience for ICLR) is very much focused on competitive results." In particular, note the first paragraph:
> > >
> > > "Does the paper present substantively new ideas or explore an underexplored or highly novel question? Papers that take risks and study a less explored area are likely to have less polished results, papers that study a highly explored topic are likely to have more polished results. This phenomenon often results in reviewers excessively penalizing papers that explore underexplored topics, and it is worth accounting for this."
> > >
> > > We study an unexplored area: task-agnostic, continual meta-learning. We sincerely believe that "a substantial fraction of the ICLR attendees [would] be interested in reading this paper" because of this particular problem setting, and our approach to addressing it. Regrettably, the reviewer did not refer to our contribution anywhere in the original review nor in the follow-up response. We encourage the reviewer to more carefully consider the potential contributions of a work when reviewing in the future.
> > >
> > > References
> > >
> > > [Gid18] Spyros Gidaris and Nikos Komodakis. Dynamic few-shot visual learning without forgetting. In Proceedings of the IEEE Computer Society Conference on Computer Vision and Pattern Recognition (CVPR). URL https://arxiv.org/abs/1804.09458.
> > >
> > > [Mun18a] Tsendsuren Munkhdalai and Adam Trischler. Metalearning with hebbian fast weights. arXiv preprint arXiv:1807.05076, 2018
> > >
> > > [Mun18b] T. Munkhdalai, X. Yuan, S. Mehri, and A. Trischler. Rapid adaptation with conditionally shifted neurons. In ICML, 2018.
> > >
> > > [Ore18] Boris N Oreshkin, Alexandre Lacoste, and Pau Rodriguez. Tadam: Task dependent adaptive metric for improved few-shot learning. In  Advances in Neural Information Processing Systems (NIPS), 2018. URL https://arxiv.org/abs/1805.10123.
> > >
> > > [Rus18] Andrei A Rusu, Dushyant Rao, Jakub Sygnowski, Oriol Vinyals, Razvan Pascanu, Simon Osindero, and Raia Hadsell. Meta-learning with latent embedding optimization. arXiv preprint arXiv:1807.05960, 2018.

---

> ### Author Response · Authors · 2018-11-22
> **[2/4] Thank you for your detailed feedback! Please let us know if there is more we can do to address your concerns.**
>
> > "There is also a nonparametric version, based on Dirichlet process mixtures, but a large number of approximations render this somewhat heuristic."
> > "Nonparametric extension via Dirichlet process mixture. This is quite elaborate, and uses further approximations (ICM, instead of Gibbs sampling). Can be seen as a heuristic to evolve the number of components.”
>
> tl;dr: Our approach is not a heuristic, but represents an alternative inference procedure that has an established history in latent variable modelling. Each method (including ICM and Gibbs) has its own trade-offs, so we do not view our selection of ICM as a detraction, nor do we claim to resolve the trade-offs between these inference techniques.
>
> While we do not estimate the full Bayesian posterior as is done with Gibbs sampling, our approach is in line with recent literature on approximate inference [Bro13, Roy13, Wan15]. In our paper, iterated conditional modes (ICM) [Bes86, Zha01, Wel06] is an established greedy strategy for iterative local maximization with guaranteed convergence (the same convergence guarantee that expectation maximization (EM) gives). In particular, ICM iteratively maximizes the full conditional distribution for each variable, instead of sampling from the conditional as is done in Gibbs sampling [Bes86]. Intuitively, it can be viewed as a special case of the framework of variational Bayes (VB) where the expectation over hidden variables (in our case, the task-specific parameters) is replaced with maximization (which can be realized as VB by taking the variational distribution as the Dirac delta distribution induced by maximization) [Wel06].
>
> Accordingly, ICM for DPMM is simply a deterministic point-estimation approximation to the same inference problem typically solved by Gibbs sampling [Wel06, Ray16]. This is line with (but slightly different from) the more recent small-variance asymptotics (SVA) derivation of Bayesian nonparametrics introduced by Kulis & Jordan [Kul12], who also derive a deterministic alternative to Gibbs. Our gradient-based optimization approach to ICM crucially alleviates requirements for conjugacy (which is difficult to uphold in our general setting of placing a hierarchical prior over a black-box estimator such as a neural network model), or the requirement to fit a variational distribution (which has its own drawbacks in terms of bias, design of the variational family, etc.).
>
> We thank the reviewer for bringing it to our attention that the tradeoffs inherent to the use of some of these methods are not clear in the current draft of the paper. For example, ICM is the most computationally efficient, while VB and Gibbs are the least efficient. Gibbs estimates are unbiased but of high variance, while all other methods potentially produce biased estimates. We do not intend to resolve these known, inherent tradeoffs, and so do not view it as a detraction of our method that we selected ICM over Gibbs sampling. Moreover, we made this selection primarily due to the computational efficiency of ICM as well as the previous interpretation of MAML as hierarchical Bayes (HB) [Gra18], which can be formalized as an ICM procedure. We will update the paper to make these tradeoffs clearer, and ask the reviewer if there remain any additional concerns over the use of ICM.
>
>
> > "These results are not near the state-of-the-art anymore, and some of the state-of-art methods are simpler and faster than even MAML."
>
> The reviewer did not provide a complete list of references for state-of-the-art methods that are "simpler and faster" so we have done our best to compile a list of possible methods. We list them in the "response to all reviewers" and detail a direct comparison with our method with attention paid to simplicity and efficiency. Please let us know if there are other methods that we should attend to.

---

> ### Author Response · Authors · 2018-11-22
> **[3/4] Thank you for your detailed feedback! Please let us know if there is more we can do to address your concerns.**
>
> > "In the case of infinite mixtures, it is not clear what is done in the end in the experiments."
> > "You use a set of size N+M per task update. In your 5-way, 1-shot experiments, what is N and M? I'd guess N=5 (1 shot per class), but what is M? If N+M > 5, then I wonder why results are branded as 5-way, 1-shot, which to mean means that each update can use exactly 5 labeled points. Please just be exact in the main paper about what you do, and what main competitors do, in particular about the number of points to use in each task update."
>
> We apologize for lack of clarity: Due to space constraints, we omitted the details about the task setup as we abided by the standard setup established by prior meta-learning literature [Vin16, Fin17]. We followed Vinyals et al. [Vin16] for the miniImageNet experiments (N=1 training datapoint and M=15 validation data points) with a meta-batch of 4 tasks. As for the regression experiments, we included in our experimental setup, at submission time, a breakdown of the mini-batch size (25) with an equal number of shots for training and validation (10) in a similar fashion to [Fin17]. We will take care to fit these domain-specific details into the next draft of the submission.
>
>
> > “What is given in Algorithm 2, is not compatible with Section 4. How do you merge your Section 4 algorithm with stochastic EM? In Algorithm 2, how do you avoid that there is always one more (L -> L+1) components? Some threshold must be applied somewhere."
>
> We did indeed use the suggested threshold in our Algorithm 2 as further detailed in Subroutine 4 in the paper at submission time. Due to space constraints, we opted to consolidate our parametric and nonparametric algorithms into a single algorithm block. This might have caused some confusion and distracted from the use of subroutines to solve crucial parts of Algorithm 2 separately for of the parametric and nonparametric variants.
>
> In the non-parametric case, the E-step (where the decision on adding a new component is made) is detailed in Subroutine 4 where a threshold (similar to the one suggested by the reviewer) is employed. In Section 5, we justify the use of such a threshold and refer to prior work that uses a similar approach. We thank the reviewer for pointing out these points of ill clarity and will revise the paper accordingly.
>
>
> > “An alternative would be to use split&merge heuristics for EM.”
>
> We would first like to clarify a misconception about our experimental setup: We consider the stochastic setting, where task assignments from previous batches are not kept in memory. Our justification for this problem setup is that it is most aligned with (and therefore easily comparable to) previous methods that make use of mini-batch optimization (including MAML, Prototypical Networks, etc.) and it is most straightforwardly adapted to the online setting in which previous data may never be revisited. An additional practical reason for the stochastic setting is that it would be extremely expensive to store the assignments for an entire dataset of the size of miniImageNet in memory. Preserving task assignments is also potentially harmful due to stale parameter values since the task assignments in our framework are meant to be easily reconstructed on-the-fly using the E-step with updated parameters \theta.
>
> As a consequence of our treatment of the stochastic setting, split-and-merge heuristics are not compatible with our approach. We cannot, for example, split a component into two, as that implies re-assigning the atoms corresponding to past tasks (which requires access to the data itself) and inferring the new component parameters based on those assignments. Merging is also not fully realizable in this setting, as it also requires re-assigning atoms.
>
> We did explore a naive approach to merging in which we computed a weighted average of the parameters of the merged components with weights proportional to the cluster sizes (realized as a moving average of data assigned to each component). However, we found the network parameter initializations resulting from this simplified approach to achieve worse validation loss. While it would certainly be interesting to consider more sophisticated heuristics for a moving summary of the task assignments (e.g., [Gom08], or a memoized approach similar to [Hug13]), we do not explore them here, since we believe our work already has sufficient novelty as the first foray into task-agnostic online meta-learning.

---

> ### Author Response · Authors · 2018-11-22
> **[4/4] Thank you for your detailed feedback! Please let us know if there is more we can do to address your concerns.**
>
> > "Results reported in Section 5 are potentially interesting, but entirely lack a reference point. The first is artificial, and surely does not need an algorithm of this complexity."
>
> We concur--the toy regression experiment was meant to be explanatory rather than to serve as an extensive experimental benchmark (see similar explanatory figures in [Fin17, Lee18] that we have found to be helpful in understanding the corresponding methods). In particular, we included this section so the reader may observe, in a simplified setting, the qualitative difference between the types of meta-learning benchmarks studied in the past and the more heterogeneous and non-stationary variants that we focus on. We expect that the toy regression setting makes the task shift especially clear, as learning to regress to the output of a convex function (such as a parabola) is quite different from learning to regress to the output of a periodic one (such as a sinusoid).
>
> We would also like to refer the reviewer to the “response to all reviewers” where we explain that our evolving dataset setting adds a previously unaddressed dimension to the standard benchmarks. It additionally proposes a crucial challenge that has been overlooked by the continual learning literature: task-agnostic continual learning. Accordingly, an algorithm of this complexity is necessary to detect shifts in the task distribution and adjust the model capacity accordingly, in contrast to current continual learning algorithms, which rely on external information about the start and end of each task as well as the number of tasks.
>
>
> > "The setup in Section 5.2 is potentially interesting, but needs more work, in particular a proper comparison to related work.  This type of effort is needed to motivate an extension of MAML which makes everything quite a bit more expensive, and lacks behind the state-of-art, which uses amortized inference networks (Versa, neural processes) rather than gradient-based."
>
> We have provided a more thorough explanatory comparison with related methods on the benchmark in Section 5.2 with related work, detailed in the "response to all reviewers" above. Additionally, we will subsequently follow up with more quantitative results, but ask for the reviewer's patience since we are restricted in terms of available computational resources at an academic lab.
>
>
> References
> ---------------
>
> [Bes86] Besag, Julian. "On the statistical analysis of dirty pictures." Journal of the Royal Statistical Society. Series B (Methodological) (1986): 259-302.
>
> [Bro13] Broderick, Tamara, et al. "Streaming variational Bayes." NeuRIPS, 2013.
>
> [Fin17] Finn, Chelsea, Pieter Abbeel, and Sergey Levine. "Model-agnostic meta-learning for fast adaptation of deep networks." ICML, 2017
>
> [Gom08] Gomes, Ryan, Max Welling, and Pietro Perona. "Incremental learning of nonparametric Bayesian mixture models." In Computer Vision and Pattern Recognition, 2008. CVPR 2008. IEEE Conference on, pp. 1-8. IEEE, 2008.
>
> [Hug13] Hughes, Michael C., and Erik Sudderth. "Memoized online variational inference for Dirichlet process mixture models." NeurIPS, 2013.
>
> [Kul12] Kulis and Jordan, "Revisiting k-means: New Algorithms via Bayesian Nonparametrics." ICML 2012.
>
> [Lee18] Lee, Y. & Choi, S.. (2018). "Gradient-Based Meta-Learning with Learned Layerwise Metric and Subspace." ICML, 2018.
>
> [Ray16] Raykov, Yordan P., et al. "What to do when K-means clustering fails: a simple yet principled alternative algorithm." PloS one 11.9 (2016).
>
> [Roy13] Roychowdhury, Anirban, Ke Jiang, and Brian Kulis. "Small-variance asymptotics for hidden Markov models." NeurIPS, 2013.
>
> [Vin16] Vinyals, Oriol, et al. "Matching networks for one shot learning." NeurIPS, 2016.
>
> [Wan15] Wang, Yining, and Jun Zhu. "DP-space: Bayesian nonparametric subspace clustering with small-variance asymptotics." ICML, 2015.
>
> [Wel06] Welling, Max, and Kenichi Kurihara. "Bayesian K-means as a maximization-expectation algorithm." ICDM, 2006.
>
> [Zha01] Zhang, Yongyue, Michael Brady, and Stephen Smith. "Segmentation of brain MR images through a hidden Markov random field model and the expectation-maximization algorithm." IEEE transactions on medical imaging 20.1 (2001): 45-57.

---

### Official Review · AnonReviewer2 · 2018-11-01
**Promising, but more work needed**

**Rating:** 4
**Confidence:** 3

**Review:**

This paper proposes a mixture of MAMLs (Finn et al., 2017) by exploiting the interpretation of MAML as a hierarchical Bayesian model (Grant et al. 2018). They propose an EM algorithm for joint training of parameter initializations and assignment of tasks to initializations. They further propose a non-parametric approach to dynamically increase the capacity of the meta learner in continual learning problems. The proposed method is tested in a few-shot learning setup on miniImagenet, on a synthetic continual learning problem, and an evolutionary version of miniImagenet.

[Strengths]

+ Modeling the initialization space is an open research question and the authors make a sound proposal to tackle this.
+ The extension to continual learning is particularly interesting, as current methods for avoiding catastrophic forgetting. inevitably saturate model parameters. By dynamically increasing the meta-learner's capacity, this approach can in principle bypass catastrophic forgetting.

[Weaknesses]

- There is nothing in the algorithm that prevents mode collapse, and the only thing breaking symmetry is random initialization. In fact, figure 5 and 6 suggest mode collapse occurs even in the non-parametric case. A closely related paper that may be of interest ( Kim et al., 2018, https://arxiv.org/abs/1806.03836 ) address this issue by using Stein Variational SGD.
- Results on miniImagenet are not encouraging; the gains on MAML are small and similar methods that generalize MAML (Kim et al., 2018, Rusu et al., 2018) achieve significantly better performance.
- Experiments on evolving tasks suggest the method is not able to capture task diversity. In the synthetic experiment (figure 5), the model suffers mode collapse when a sufficiently difficult task is introduced. Ultimately, it performs on par with MAML, despite having three times the capacity. Similarly, on the evolving miniImagenet dataset, figure 6 indicates there is no cluster differentiation across tasks.
- The paper needs major polishing.

---

> ### Author Response · Authors · 2018-11-22
> **[1/2] Thank you for your feedback! Please let us know if there is more we can do to address your concerns.**
>
> We thank the reviewer for their comments. We respond below to specific comments below but please also see the general "response to all reviewers" above.
>
>
> > "Results on miniImagenet are not encouraging; the gains on MAML are small and similar methods that generalize MAML (Kim et al., 2018, Rusu et al., 2018) achieve significantly better performance."
>
> For the standard homogeneous miniImageNet benchmark, we would first like to refer the reviewer to the “response to all reviewers” where we emphasize that our primary goal is not necessarily to achieve state-of-the-art results on these traditional datasets, and, moreover, benchmarking on this dataset is difficult due to nonstandard practices.
>
> However, as reported in the paper at submission time, our model does achieve the highest 1-shot accuracy for comparable architectures. The reported higher accuracies in the lower half of Table 2 use different and significantly more powerful architectures.
>
>
> > "There is nothing in the algorithm that prevents mode collapse, and the only thing breaking symmetry is random initialization… A closely related paper that may be of interest ( Kim et al., 2018, https://arxiv.org/abs/1806.03836 ) address this issue by using Stein Variational SGD."
>
> tl;dr: Auxiliary mode collapse penalties (analogous to the repulsion term in BMAML [Kim18]) might not be appropriate for clustering in the stochastic setting.
>
> Regarding the use of Stein Variational Gradient Descent in Kim et al. [Kim2018]: The second term in Eq. (1) represents a repulsive force which might deter mode collapse to some degree. However, their approach does not necessarily handle multimodality in the case of heterogeneous tasks better than our proposed approach with a similar number of particles (to our number of components), as a small number of particles could still concentrate around one large mode and ignore the narrower ones. In particular, their repulsion term does not guarantee differentiation, nor do they investigate whether the phenomenon of mode collapse occurs in their experiments (either with the repulsion term or with an ablation of the repulsion term).
>
> Regarding our method: We confirm the reviewer's assessment that symmetry-breaking in the method described in the submission is only due to the random seeding of the cluster initializations. Using random initialization alone to break symmetry is a common practice in the clustering and latent mixture modelling literature due to its simplicity [e.g., Pen99]; more sophisticated approaches, such as data-dependent initialization schemes, would be orthogonal to our approach.
>
> A complication regarding imposing auxiliary regularization during training to break symmetries is that we are working in the stochastic setting, where task assignments from previous batches are not kept in memory; therefore, any such regularization terms must be evaluated per batch. However, it is difficult to impose a principled, batch-wise regularization term that encourages mode differentiation without making assumptions about the task distribution within a mini-batch. Since our meta-learning training formulation assumes tasks are sampled uniformly with replacement from a potentially non-stationary task distribution, it would be disadvantageous in the general case to make such assumptions.
>
> In particular, an artificial penalty to enforce differentiation of assignments within a batch could hinder information transfer between two (somewhat similar) tasks by forcing their assignments into two different clusters. This assumption also crucially falls apart in the case of evolutionary miniImagenet (Figure 6), where the tasks in a batch do share the same stylization, and therefore may benefit from being assigned to the same cluster. For these reasons, although it is straightforward to include a batch-wise regularization term (one that, for example, penalizes the entropy of a categorical distribution, which would be analogous to the batch-wise repulsion term used in BMAML), we do not believe that this is appropriate for the general problem setting that we consider. Ideally, we want only to enforce/diminish transfer between tasks that share/lack similar properties, which is realized via our underlying probabilistic model.

---

> ### Author Response · Authors · 2018-11-22
> **[2/2] Thank you for your feedback! Please let us know if there is more we can do to address your concerns.**
>
> > "In fact, figure 5 and 6 suggest mode collapse occurs even in the non-parametric case."
> > “Ultimately, it performs on par with MAML, despite having three times the capacity”.
>
> We apologize for the confusion caused by the original version of this figure. Notably, what we represent in Figure 5 is the validation loss values for each task, on a logarithmic scale. Accordingly, Figure 5 confirms that our model presents a substantial improvement over MAML that justifies the added complexity. Moreover, there is no mode collapse in the synthetic regression experiments, since Figure 5 shows that the spawned clusters were sufficiently differentiated (and at most one type of task was assigned per component).
>
> We will add a table with the final loss values for a clearer comparison in an updated PDF submission.
>
>
> > “Experiments on evolving tasks suggest the method is not able to capture task diversity... Similarly, on the evolving miniImagenet dataset, figure 6 indicates there is no cluster differentiation across tasks”
>
> We would like to emphasize that Figure 6, as well as Figure 5, do demonstrate task differentiation to a reasonable degree. Note that the total cluster responsibility reported in Figure 6 is the sum of cluster responsibilities across the different tasks in a single mini-batch. Accordingly, at each moment, one or two clusters are assigned tasks (from the minibatch of 4 tasks) with a non-zero probability.
>
> In a later version, we will present two figures for each experiment, one for the losses and one for the cluster responsibilities, to avoid further confusion. We are also working on a more visually informative and less overwhelming presentation of the cluster assignment probabilities per task to emphasize the capability of our approach to differentiate between tasks and spawn new clusters when needed, in a task-agnostic setting.
>
>
> > "The paper needs major polishing."
>
> Thank you for bringing this up. We have devoted significant effort to increase clarity in the revised version.
>
>
> References
> ------------
>
> [Kim18] Kim, Taesup, et al. "Bayesian Model-Agnostic Meta-Learning." NeurIPS, 2018.
>
> [Pen99] Pena, José M., Jose Antonio Lozano, and Pedro Larranaga. "An empirical comparison of four initialization methods for the k-means algorithm." Pattern recognition letters 20.10 (1999): 1027-1040.

---

> ### Comment · AnonReviewer2 · 2018-11-26
> **Concerning miniImagenet**
>
> Dear authors,
>
> Thank you for an extensive rebuttal. Your frustration is palpable, but I do not believe my position on the miniImagenet experiment, which overlaps largely with R1, is unfair.
>
> You dedicate considerable effort to defend you results on miniImagenet, primarily on the grounds that (a) the purpose is not to match state of the art (b) your architecture is less powerful and should not be compared against SOTA, and (c) differences in implementation invalidates direct comparisons. These are reasonable points, but you miss my key concern, elaborated below.
>
> I agree that there are several challenges with miniImagenet benchmarks. That places an additional burden on any paper aiming to include such benchmarks, especially in terms of reporting experimental setup, discussing what comparisons can be made, and which results are incompatible. Your current manuscript mentions miniImagenet twice; in the abstract and table 1. There are no experimental details, no motivation for including this experiment, nor any discussion of the results. You claim this is due to page count, but you have 1.5 pages to the max limit and a largely empty appendix. A consequence of this choice is that a reader is forced to be maximally conservative (i.e., compare against SOTA).
>
> Given the authors rebuttal, I wonder what the purpose of this experiment is. If it is to evaluate the parametric mixture model, I am missing a carefully controlled experiment highlighting how performance varies with the number of components, or at the very least a suggestion for a relevant baseline that results can be compared against.
>
> In particular, comparing against MAML alone is problematic: since your approach is a mixture of MAMLs it is essentially guaranteed to do at least as well. The only interesting question is *how much* of an improvement you can realize, and what is driving it. Being a mixture of MAMLs, you have a confounding factor in the increased capacity of the meta learner. It is therefore impossible for a reader to discern from table 1 whether the performance gain is due to your mixture model or simply due to increased capacity. One option would have been to control for confounders through an ablation study. In the absence of that, relevant comparisons must be found in competing methods attempting to capture heterogeneity in the data. Again, this is not a requirement to outperform SOTA, but failing to compare favourably against similar methods does not provide evidence in favor of yours.
>
> One such benchmark that the authors discuss in the rebuttal is BMAML (Kim et. al, 2018). I agree that BMAML attempts to solve a slightly different problem; even so, it is trying to capture heterogeneity and has strong analogues to your method in that it maintains a set of parameterizations to capture multi-modality. At the very least, it provides an interesting comparison that would have been beneficial to highlight in the paper. As you point out, Kim et al. (2018) use a baseline MAML with higher performance than reported in the original paper (Finn et al., 2017) (likely driven the use of 5 convolutional layers, as opposed to 4). Even so, they carefully control for the effect of increasing the number of particles and show that going from 1 (equivalent to MAML) to 3 or 5 particles yields a 3-percentage point increase in performance. This is a relative measure on a comparable architecture, and as such does provide a relevant baseline. In contrast, the current manuscript reports a 2-percentage point increase in performance over a baseline MAML without controlling for (or even reporting) the number of components in the mixture. BMAML is just one possible baseline, others that aim to capture heterogeneity using a comparable architecture (e.g. Gidaris & Komodakis, 2018) realize even greater performance gains.
>
> Taken together, the authors conduct an experiment without context or discussion, and as such fail to provide any insights into the workings of the method or validate claims made. The fact that the method does not hold up against similar (in some respect) methods must be interpreted as evidence, if not against, at least not in favour of the proposed method.

---

> ### Comment · AnonReviewer2 · 2018-11-26
> **Concerning review-specific rebuttal (1/2)**
>
> Dear authors,
>
> These comments elaborate on concerns I raised in my initial review. Let me first emphasize that my evaluation is not primarily driven by the miniImagenet experiment, but what I maintain is evidence of either mode collapse or catastrophic forgetting. Below, I respond to your specific comments.
>
> > tl;dr: Auxiliary mode collapse penalties (analogous to the repulsion term in BMAML [Kim18]) might not be appropriate for clustering in the stochastic setting.
>
> I agree that auxiliary penalties are a non-trivial problem and any approach comes with limitations. You are right that the repulsive force does not guarantee the avoidance of mode collapse, but certainly, it will do more than having no repulsion. I’m not sure that just because random initialization breaks symmetry in the standard use case, it will do so when applied over a distribution of gradient fields. To understand my concern, note that all your parameters are initialized around origin (presumably); from that point, the variation in initialization may very well be too small to prevent all gradients from pointing in the same direction and does so only if the loss surface is hyper-sensitive. This may be a valid assumption in high-dimensional space–or it may not. Notably, MAML, and as such a mixture thereof, is not scale invariant. If one task has gradients an order of magnitude larger than any other, all mixture components will be dragged towards that (or some) task-specific minima.
>
> The current manuscript is an attempt to tackle perhaps too much; with a known task distribution you can leverage more information in your objective and could possibly encourage greater simultaneous exploration of initialization space. Alternatively, focusing on the non-stationary continuous learning setup could motivate certain mechanisms for avoiding catastrophic forgetting. In the current manuscript my main concerned is that I see no strong evidence suggesting you achieve either.
>
> > We apologize for the confusion caused by the original version of this figure. Notably, what we represent in Figure 5 is the validation loss values for each task, on a logarithmic scale.
>
> First note that only the third panel is on a logarithmic scale. Further note that the caption does match the figure, and the description of the results in the main text agree with neither. As such, I may very well have misinterpreted these results. Thus, let me be very specific.
>
> - First, I note that the caption states “We plot the negative log likelihood of the data across all tasks under each cluster”. This makes no sense for the baseline MAML, which only has one “cluster”. Since the curves for MAML differs between panels, and reduction in loss generally agree with when new tasks are being introduced, my interpretation is that each panel plots the validation loss for a specific task. Your rebuttal confirms this interpretation. I surmise the top panel is task 1, the middle panel task 2, and the bottom panel task 3. In the case of MAML, there is no cluster differentiation between panels.
>
> - Second, I note that both models do equally well on task 1 until step ~700 (after all, they are equivalent at this point). After that, task 2 is introduced and the baseline’s performance on task 1 deteriorates. It also fails to learn the closely related task 2, suggesting it might be poorly calibrated. In contrast, the mixture model retains performance on task 1 and learns task 2.
>
> - at ~1500 (not 2100, as claimed), the validation loss on task 3 goes down for both the baseline and the mixture. The mixture model learns the new task faster, but MAML again appears to behave oddly. More importantly, the mixture model fails to retain performance on the two previous tasks, and from step 1500 to 2500 validation losses on task 1 and 2 increases to a level indicating a catastrophic loss of performance.
>
> My suspicion is that this happens because the loss with respect to the third task is many orders of magnitude larger. Thus, my concern with respect to mode collapse kicks in; every component in the mixture will receive very large gradient updates pulling them towards a local minimum on task 3, effectively collapsing into, if not a single mode, modes useful for task 3 only. In other words, the mixture model is suffering catastrophic forgetting over a sequence of three tasks. This makes me worried what would happen over a longer sequence, as is usually seen in continual learning, and in a more complex environment.

---

> > ### Author Response · Authors · 2018-12-05
> > **Thank you for your response. [1/3]**
> >
> >  We thank the reviewer for the response. We respond to specific points below.
> >
> > >  "...with a known task distribution you can leverage more information in your objective and could possibly encourage greater simultaneous exploration of initialization space."
> >
> > We reiterate that the task-agnostic setting is the core problem that our approach attempts to address, and therefore respectfully disagree that the setting is beyond scope. It does indeed present additional challenges but represents a more setting that is appropriate for applications in which the underlying task distribution has an uncontrollable, non-stationary component.
> >
> > > "Notably, MAML, and as such a mixture thereof, is not scale invariant. If one task has gradients an order of magnitude larger than any other, all mixture components will be dragged towards that (or some) task-specific minima."
> > > "My suspicion is that this happens because the loss with respect to the third task is many orders of magnitude larger."
> >
> > The reviewer is correct: The regression loss is unbounded and differences in magnitude between types of tasks (e.g., sinusoidal vs. polynomial) may be significant. Continual learning is difficult for any method that makes use of the loss on each type of task as a measure of catastrophic forgetting. A possible, but not immediate, solution is the normalization of loss scale between tasks. However, in our task-agnostic setting, such normalization would require prior information about the task distribution, which we do not assume access to.
> >
> > We note that other established methods for continual learning, such as elastic weight consolidation (EWC) [Kir2017], synaptic intelligence (SI) [Zen17], and variational continual learning (VCL) [Ngu17], are also subject to the concern of incomparable task losses, as they make use of the loss for each task in order to overcome catastrophic forgetting. Moreover, although these methods are task-aware, we are not aware of an intrinsic algorithmic component that achieves normalized loss scales. We believe that this is because it is difficult to disentangle catastrophic forgetting from the inherent difficulty of a task simply by looking at the model's performance on the task (as measured by the task loss). Such an algorithmic component would be an interesting contribution, but is one that is not addressed in these works, nor is it one that we intend to address here, although we believe an approach such as gradient normalization (https://arxiv.org/abs/1711.02257) may be promising.
> >
> > We acknowledge that we did not clarify these details surrounding loss scaling in the submission, and, as such, the lack of robustness to differences in loss scale between tasks could seem unique to our method. We also note that this issue is not prevalent in the standard benchmarks for continual learning (e.g., permuted MNIST), since these tasks are dealt with using the cross-entropy error, and thus the loss scaling is not significantly different across tasks. We thank the reviewer for pointing out this unrepresentative choice of tasks for the synthetic regression benchmark; however, we will defer an appropriate revision to a future draft of this work, as we could not revise this section in the day between the reviewer's comments and the revision deadline.

---

> > ### Author Response · Authors · 2018-12-05
> > **Thank you for your response. [2/3]**
> >
> > > "I’m not sure that just because random initialization breaks symmetry in the standard use case, it will do so when applied over a distribution of gradient fields ... your parameters are initialized around origin (presumably); from that point, the variation in initialization may very well be too small to prevent all gradients from pointing in the same direction and does so only if the loss surface is hyper-sensitive. "
> >
> > We note that we are not explicitly maintaining a distribution over gradient fields. However, the reviewer is correct in identifying another component on which we could perform an ablation: The random per-component initialization. We will perform the corresponding ablation that sets a new mixture component to the same value as another component.
> >
> > We note, however, that too small a variance in initialization is not a concern of our approach. In particular, the spawning of a new cluster makes use of a variance hyperparameter (a parameter of the global prior) that we tune to ensure initial differentiation without having useless clusters. High-dimensionality may or may not play a role in differentiation at the initialization point, but this is a matter of empirical verification that is not within the scope of our paper.
> >
> > There is, however, another component that breaks symmetry later on in training, as more data has been observed: the richer-get-richer property of predictive distribution for a cluster identity in Bayesian nonparametric clustering. As a consequence, the probability of joining an existing cluster is proportional to the size of that cluster. Over the course of training, the rich-get-richer property ensures that the meta-parameters do *not* follow the same optimization trajectory, and is thus another component driving specialization.
> >
> > >  regarding synthetic experiments:
> > The way in which we initialize a new cluster plays a role in the capacity of the mixture to succeed on a new task as compared to the baseline MAML. The new cluster is initialized with a sample from the global prior (i.e., it is perturbed). This perturbation may aid adaptation to a novel task. In contrast, the MAML parameters are applied without change to the new task data.
> >
> > > Comparison to particle methods with repulsion terms:
> > We re-iterate that particle methods such as B-MAML, while they have the potential to capture multiple modes, are not optimal for the evolving setting. Furthermore, our interest is in clustering similar tasks together for optimal transfer. Particle methods do not perform that function, as they cannot constrain the meta-gradient update based on the cluster assignments of the tasks. In particular, note the lack of weighting (e.g., by a factor related to the repulsion term) in the meta-loss in Eq. (5) of BMAML [Kim18].
> >
> > > "[the authors of BMAML] show that going from 1 (equivalent to MAML) to 3 or 5 particles yields a 3-percentage point increase in performance. This is a relative measure on a comparable architecture, and as such does provide a relevant baseline."
> > We reported a 2% improvement with 5 components, in contrast to the 3% improvement of BMAML with 5 particles. However, given that the architectures are **not** comparable (BMAML: 5 layers of 64 filters; ours: 4 layers of 32 filters), this contrast is inconclusive, as more parameters might ease differentiation. As we stated in the main revision report, we are happy to run a more standardized comparison, but cannot straightforwardly do so as the authors of BMAML have not to our knowledge released code.
> >
> > > “First note that only the third panel … differentiation between panels”:
> > We apologize for the confusion caused by the mismatch between the figure and the main text due to a mixup at submission time. We have fixed that issue in addition to fixing the outdated caption in the submitted revision.
> >
> > > “After that, task 2 is introduced … might be poorly calibrated. at ~1500 … appears to behave oddly”
> > Here, we believe that MAML, at the end of the first training phase, converged to a local minima with respect to the second task distribution loss. Accordingly, MAML updates did not successfully learn the slightly different distribution of tasks (odd vs. even polynomial regression). In contrast, our algorithm spawns a new cluster that is further from such a stationary point (due to the perturbation induced by the global prior G) and is thus able to better learn the new tasks. However, we understand that the current figure might appear to report a subpar choice of a MAML run; in a future revision of the paper, we will rerun this experiment for a larger number of times to plot the mean and variance/CI for the validation loss at each step of the training process to investigate this phenomenon.

---

> > ### Author Response · Authors · 2018-12-05
> > **Thank you for your response. [3/3]**
> >
> > > “More importantly, the mixture model fails to retain performance on the two previous tasks ... indicating a catastrophic loss of performance.”
> > As there is no standard threshold for what qualifies as “catastrophic forgetting”, we quantitatively compare to MAML. MAML indicates a more significant loss of performance. It is quite likely that the different loss scales across task distributions is the main cause of the decay in performance for our method (as discussed in detail above in this response). However, the final validation loss on each task is not far from the best loss (at the end of each corresponding training phase), and, importantly, is superior to MAML. If the reviewer has any suggestions for a more appropriate baseline than MAML in this setting, we would very much appreciate the suggestion.
> >
> > > “when a new task is introduced, *three* clusters are immediately created … suggesting one of them is redundant despite there now being three tasks.”
> > The unintended behavior seen in this figure is the unnecessary spawning of a 3rd cluster right after the 2nd one. As such, *two* (not three) clusters were spawned *almost at the same time* (within 200 iterations of each other, not immediately). Understandably, there was not much differentiation between these two clusters (making one of them redundant) for the rest of this training phase. We have, however, found better cluster differentiation in more recent runs of our algorithm (with a different set of hyperparameter values) that we will add to a later revision of this work. We have moved the current miniImageNet figure to the appendix.
> >
> > > “Without plotting the actual assignment distribution, it is hard to say anything..."
> > We agree with the reviewer that the unnormalized plot of validation negative log likelihoods would not strongly demonstrate our point regarding task differentiation. However, we would like to refer the reviewer to a new figure (Fig. 6) for the regression setting where we demonstrate sufficient differentiation across tasks and clusters by plotting the assignment distribution. We would, however, like to clarify that the change of task distribution between the training phases does not guarantee that the distributions have zero overlap. Accordingly, we cannot expect the clusters to be perfectly specialized.
> >
> > > “small batch sizes (not reported in the manuscript)”
> > > “doesn't explain the sudden change in the middle of training on a given task”
> > We refer the reviewer to the “hyperparameters” paragraph of the corresponding experiment, where the batch size is reported. Given the potentially high variance of the gradient of the validation loss across episodes due to the small batch of training data (1 example per class) as well as the relatively small meta-batch size (4), it is not surprising at all to see noisy fluctuations in parameter values. However, an additional source of noise in these diagrams is the small batch size we have used when evaluating on the meta-validation data. We will correct this by reporting the average validation loss on the entire meta-validation dataset for parameters at each iteration interval of training; this is costly but gives a much more stable estimate of generalization.
> >
> > > “A more general point on both these experiments is that a sequence of three task is a very short sequence...”
> > We based our sequence of 3 on the permuted MNIST experiments in the Elastic Weight Consolidation paper [Kir2017]. However, we agree that we could better demonstrate our point with a longer sequence, and will explore such experiments in a future version of this work.
> >
> > > “Further, introducing new experiments...without a relevant baseline makes your results hard to relate to.”
> > We would like to note that, at the time of submission, there had been *no prior work on continual meta-learning*; the fact that MAML will trivially forget catastrophically is only another justification to the urgent need for a method that can tackle the non-stationary meta-learning setting. To dismiss this work’s contribution based on how a standard method for meta-learning cannot handle an evolving setting is analogous to dismissing the whole of the continual learning literature because SGD trivially forgets catastrophically.
> >
> > We have considered different datasets to demonstrate our method, however, we could not find suitable datasets or sets of tasks suitable for *both “evolving” and “few-shot”*. Note that the continual learning datasets the reviewer mentioned (permuted MNIST/CIFAR-10) are not immediately suitable for few-shot learning or meta-learning as there is no standardized batching into task episodes, each with a train and validation batch. Moreover, stylized miniImageNet is much more complex than these datasets (in terms of both pixel density and image content). Previous approaches to continual learning have never, to the best of our knowledge, been applied to data of this complexity.

---

> ### Comment · AnonReviewer2 · 2018-11-26
> **Concern review-specific rebuttal (2/2)**
>
> > Note that the total cluster responsibility reported in Figure 6 is the sum of cluster responsibilities across the different tasks in a single mini-batch. Accordingly, at each moment, one or two clusters are assigned tasks (from the minibatch of 4 tasks) with a non-zero probability.
>
> Admittedly, I found this figure extremely difficult to interpret and am still confused by it. It appears to me that after 20K iterations, when a new task is introduced, *three* clusters are immediately created, though only two should be needed. Moreover, the cluster assignment seems to fluctuate more than it should: the green and red cluster essentially take turns being assigned, which makes little sense to me. Moreover, after 25K steps (still two tasks), all three clusters seem to be active always with roughly equal responsibilities, suggesting the allocation may be largely random or at least not diversified as one would expect. Without plotting the actual assignment distribution, it is hard to say anything about how what these clusters are doing and how consistent they are with respect to the tasks. For instance, when the third task is introduced, something equally confusing happens at 35K steps. All of a sudden, green and red clusters again starts taking turns being assigned, suggesting one of them is redundant despite there now being three tasks. This may be an artifact of very small batch sizes (not reported in the manuscript), but that doesn't explain the sudden change in the middle of training on a given task.
>
> I am certainly open to the possibility of having misinterpreted this figure (and the former), if so please do correct me.
>
> A more general point on both these experiments is that a sequence of three task is a very short sequence to demonstrate an ability to modulate over an evolving task distribution. Further, introducing new experiments (there are several benchmarks on versions of MNIST or evolving Cifar10 (Zenke et al., 2017), why not use any of those?) without a relevant baseline makes your results hard to relate to. For instance, MAML is trivially going to incur catastrophic forgetting in an evolutionary setting. I do take your point though that the evolving miniImagenet experiment is mainly about exploring cluster differentiation.

---

### Official Review · AnonReviewer3 · 2018-11-05
**A more systematic evaluation is necessary**

**Rating:** 5
**Confidence:** 2

**Review:**

This paper presents a mixture of hierarchical Bayesian models for meta-learning to modulate transfer between various tasks to be learned. A non-parametric variant is also developed to capture the evolution of a task distribution over time. These are very fundamental and important problems for meta-learning. However, while the proposed model appears to be interesting, the evaluation is less convincing.

1. The performance of few-shot classification on MiniImageNet is not comparable to the state of the art (Table 2, Table 1). Especially, by Table, the proposed model performs much worse than existing methods (50% vs 60%). More discussions and explanations on this experiment are clearly required.

2. A more systematic and realistic evaluation is necessary to justify the proposed method. As a method that aims to cope with heterogeneous or even evolving task distributions, it is expected to work well in practice and outperform those baselines that are designed for a single task distribution.

---

> ### Author Response · Authors · 2018-11-22
> **Thank you for your feedback! Can you elaborate on your comments?**
>
> We thank the reviewer for their comments. We are in agreement with the reviewer that both modulating transfer and ensuring robustness to a changing task distribution are important and timely problems in meta-learning.
>
> We respond below to specific comments but please also see the general "response to all reviewers" above.
>
>
> > "The performance of few-shot classification on MiniImageNet is not comparable to the state of the art (Table 2, Table 1)... More discussions and explanations on this experiment are clearly required."
>
> For the standard homogeneous miniImageNet benchmark, we would first like to refer the reviewer to the “response to all reviewers” where we emphasize that our primary goal is not necessarily to achieve state-of-the-art results on these traditional datasets, and, moreover, benchmarking on this dataset is difficult due to nonstandard practices.
>
> However, as reported in the paper at submission time, our model does achieve the highest 1-shot accuracy for comparable architectures. The reported higher accuracies in the lower half of Table 2 use different and significantly more powerful architectures.
>
>
> > "A more systematic and realistic evaluation is necessary to justify the proposed method. As a method that aims to cope with heterogeneous or even evolving task distributions, it is expected to work well in practice and outperform those baselines that are designed for a single task distribution."
>
> We apologize for the confusion caused by the original version of this figure: Notably, what we represent in Figure 5 is the validation loss values for each task, on a logarithmic scale. Accordingly, Figure 5 confirms that our model presents a substantial improvement over MAML that justifies the added complexity of our method.
>
> We would also like to emphasize that Figure 6, as well as Figure 5, demonstrate task differentiation to a reasonable degree. Note that the total cluster responsibility reported in Figure 5 and Figure 6 is the sum of cluster responsibilities across the different tasks in a single mini-batch. Figure 5 shows that the spawned clusters were sufficiently differentiated (and at most one type of task was assigned per component). In Figure 6, at each moment, one or two clusters are assigned tasks (from the minibatch of 4 tasks).
>
> In a later version, we will add a table with the final loss values, and we will present two figures for each experiment, one for the losses and one for the cluster responsibilities, to avoid further confusion. We are also working on a more visually informative and less overwhelming presentation of the cluster assignment probabilities per task to emphasize the capability of our approach to differentiate between tasks and spawn new clusters when needed, in a task-agnostic setting.
>
> We would welcome more specific comments from the reviewer on what would constitute a more systematic and realistic evaluation.

---

### Author Response · Authors · 2018-11-22
**Response to all reviewers**

We thank the reviewers for their constructive comments. Below, we clarify some common points of concern regarding significance, originality and clarity, and hope that this response can facilitate ongoing discussion during the rebuttal period. We will subsequently follow up in an updated PDF submission with improvements in writing and presentation quality, as well as a clearer experimental comparison as some of the reviewers suggested. We would be happy to discuss any remaining concerns in the intervening time period.

We share the enthusiasm of reviewers towards tackling the problem of adaptively determining how much to transfer from previous tasks in a meta-learning setting (which we refer to in the paper as "transfer modulation") thus avoiding negative transfer and promoting positive transfer. We note that no recent prior method (including Prototypical Networks [Sne17], VERSA [Gor18], and MAML [Fin17]) explicitly proposes a solution to this challenge, nor has there been recent prior work in meta-learning that empirically investigates this phenomenon.

We also share the reviewers' enthusiasm towards developing meta-learning methods that possess adaptive complexity, as a natural continuation of recent progress in meta-learning with static datasets. As Reviewer 2 identified, current methods for meta-learning (including Prototypical Networks [Sne17], VERSA [Gor18], MAML [Fin17] and variants thereof [Kim18, Fin18]) inevitably saturate model parameters in the evolving dataset regime. Moreover, techniques developed specifically to address the catastrophic forgetting problem, such as elastic weight consolidation (EWC) [Kir2017], synaptic intelligence (SI) [Zen17], and variational continual learning (VCL) [Ngu17], require access to an explicit delineation between tasks that acts as a catalyst to grow model size. (We may refer to such methods as "task-aware.")

In contrast, our nonparametric algorithm tackles the "task-agnostic" setting of continual learning, where the meta-learner does not receive information about task changes but instead learns to recognize a shift in the task distribution and adapt accordingly. The task-agnostic setting is more realistic and inherently more difficult. Addressing this setting is a contribution of our work that we under-emphasized in the first submission but that justifies the complexity of the proposed nonparametric algorithm. We will revise the paper to make this contribution more central.

Before we address specific concerns, we would like to emphasize that our *primary* goal is not to make significant gains in the state of the art in traditional meta-learning tasks. The standard benchmarks for meta-learning, such as Omniglot and miniImageNet, were designed with a uniform distribution of tasks in mind. This assumption falls short of many scenarios in the real world, where the task distribution is significantly heterogeneous or nonstationary. A simple example would be the difficulties imposed by changes in the agent’s environment (e.g., the change from day to night for computer vision tasks, or terrain changes in the context of a locomotive controller in reinforcement learning) leading to a novel set of tasks which might be quite different from those previously encountered.

Rather, our intent is to investigate how the heterogeneity or non-stationarity of a dataset affects the performance of existing meta-learning algorithms, as well as to propose a tailored solution to these challenges. As an empirical investigation into such settings, we opted for an inventory of stylization effects in our design of a new evolving dataset derived from miniImageNet. Consequently, our results suggest that recognizing such underlying discrete structure via hierarchical modelling can improve performance via robustness to such abrupt changes. As such, we do not believe the modest improvement on the homogeneous miniImageNet task to be grounds for rejection. Moreover, as we detail below, the state of benchmarking in this domain is challenging due to nonstandard practices.

---

> ### Author Response · Authors · 2018-11-22
> **Response to all reviewers: References**
>
> References
> ---------------
>
> [Fin17] Finn, Chelsea, Pieter Abbeel, and Sergey Levine. "Model-agnostic meta-learning for fast adaptation of deep networks." ICML, 2017.
>
> [Fin18] Finn, Chelsea, Kelvin Xu, and Sergey Levine. "Probabilistic Model-Agnostic Meta-Learning." In NeurIPS, 2018.
>
> [Gar18] Garnelo, Marta, Jonathan Schwarz, Dan Rosenbaum, Fabio Viola, Danilo J. Rezende, S. M. Eslami, and Yee Whye Teh. "Neural processes." arXiv preprint arXiv:1807.01622 (2018).
>
> [Gor18] Gordon, J., Bronskill, J., Bauer, M., Nowozin, S. and Turner, R.E., 2018. "Decision-Theoretic Meta-Learning: Versatile and Efficient Amortization of Few-Shot Learning." arXiv preprint arXiv:1805.09921.
>
> [Gra18] Grant, Erin, et al. "Recasting gradient-based meta-learning as hierarchical Bayes." ICLR, 2018.
>
> [Kim18] Kim, Taesup, et al. "Bayesian Model-Agnostic Meta-Learning." NeurIPS, 2018.
>
> [Koir2017] Kirkpatrick, James et al. "Overcoming catastrophic forgetting in neural networks." In Proceedings of the national academy of sciences (2017).
>
> [Ngu17] Nguyen, Cuong V. et al. "Variational continual learning." ICLR, 2018.
>
> [Sne17] Snell, J., Swersky, K. and Zemel, R. "Prototypical networks for few-shot learning." NeurIPS, 2017.
>
> [Zen17] Zenke, Friedemann, Ben Poole, and Surya Ganguli. "Continual learning through synaptic intelligence." ICML, 2017.

---

> ### Author Response · Authors · 2018-11-22
> **Response to all reviewers: Variations in model architecture and experimental setup in recent works on meta-learning result in non-standardized benchmarking**
>
> As the reviewers have pointed out, there has been a flurry of recent papers on meta-learning and few-shot learning that perform benchmarking on the miniImageNet few-shot classification dataset. We agree that it is important to track these trends, as standardization via benchmarking is integral to validation in an empirically-driven field. However, we have found the evaluation on miniImageNet to be nonstandard, making benchmarking difficult.
>
> The reported accuracies on miniImageNet appear to be not only the result of algorithmic improvements but also changes in model architecture and experimental setup. In particular, we note that it is difficult to pinpoint the implementation differences in approaches that give rise to reported improvements. As an illustrative example, the BMAML method reports better performance than MAML [Fin17] for 1 particle, despite the fact that in this special case the method reduces to exactly MAML. Another potential complication is that the generation of the miniImageNet few-shot episodes is nonstandard, resulting in differences in the training, validation and testing datasets between papers, which certainly has an effect on the reported test accuracies.
>
> To combat these potential confounds, we have used the same data-generation procedure (seeded with the same random seed) as MAML [Fin17] to generate training, validation and testing episodes, and kept all parameters that are common to both our methods constant (including, for example, the inner loop learning rate and the network architecture). We subsequently report an improvement while adhering to these constraints, which we request the reviewer to not take lightly.
>
> Below, we give a more detailed comparison with related methods that focus on a probabilistic approach to meta-learning (which we are happy to add to if the reviewers would find it useful).
>
>
> ** Bayesian model-agnostic meta-learning (BMAML) [Kim18]
>
> This method makes use of Stein variational gradient descent (SVGD) to maintain a Monte Carlo estimate of the posterior over task-specific parameters \phi given the task-specific training data X[1:N], Y[1:N]. This is, therefore, an alternative to a MAML-like gradient-based optimization technique, which would employ a point estimate for \phi.
>
> However, a confusing result in this paper is that the baseline of 1 particle, which should perform exactly in line with the standard MAML implementation ("Because SVGD with a single particle, i.e., M = 1, is equal to gradient ascent, Algorithm 2 reduces to MAML when M = 1" pg. 4) achieves 50.60% ± 1.42% on the standard miniImageNet benchmark (cf. 48.7% ± 1.84% [Fin17]). This is an improvement of almost 2% due to an undisclosed change in the experimental setup. Because of this, as well as the fact that the code for BMAML not been released at this time, we are unable to perform a direct comparison with BMAML, and would caution against interpreting the raw difference in accuracies as conclusive evidence that our method is unfit for publication.
>
>
> We wish to emphasize again that our motivation for the proposed method is not simply to get as high performance as possible on the standard benchmark, and as such, we have tightly restricted variations in hyperparameters, such as the inner loop learning rate and neural network architecture, that could potentially improve performance. Instead, we aim to explore the analogy between gradient-based meta-learning and probabilistic modelling, which has served to inspire many of the papers that the reviewer cites, among others. Our exploration focuses on the structure of the underlying probabilistic model and how inference and parameter estimation can be performed efficiently using a MAML-like gradient-based optimization technique.
>
> Moreover, we do show an improvement on techniques in the same lineage (i.e., gradient-based meta-learning with the standard convnet architecture of [Vin16]). The methods that show further improvements consider alternative methods for inferring task-specific parameters (which is somewhat orthogonal to the structure of the underlying probabilistic model), but also, in many cases, changes to the experimental setup and the model architecture, as described in detail above.

---

> ### Author Response · Authors · 2018-11-22
> **Response to all reviewers: Alternate meta-learning algorithms correspond to the use of different inference techniques**
>
> tldr; Our intent in this paper is not to pit against each other techniques for inference of task-specific parameters (e.g., MAML vs. BMAML vs. VERSA), but to investigate how changes to the underlying probabilistic model (the hierarchical Bayesian model) can be realized as changes to procedures executed in a meta-learning algorithm. Our approach thus provides guidance in algorithm design for meta-learning. We believe that such explorations are timely and of great interest to the ICLR community.
>
> We would like to emphasize what we view as the differences between our approach and recent methods that tackle the meta-learning problem using probabilistic methods (e.g., BMAML [Kim18], VERSA [Gor18], neural processes [Gar18]). Such approaches share an assumption about the structure of the underlying probabilistic model, namely the hierarchical Bayesian model with exchangeability across tasks as well as exchangeability across data within a task, depicted in our paper in Figure 1(a). What differs among these methods, then, is not a different assumption of dependence relationships between latent variables (the task-specific parameters, \phi), but the inference procedure implemented to infer their values. (More detail given below.)
>
> In contrast, we propose a set of different structural assumptions that are appropriate for certain practical settings: When there is known to be a latent clustering structure in a dataset of tasks. This corresponds to the graphical model visualized in Figure 1(b-c). The change to the underlying probabilistic model induces some benefits that do not result from changing the inference procedure: the ability to detect changing tasks and how tasks can be grouped via inspection of the latent variables, and, importantly, a principled way to adapt the complexity of a model in response to an evolving dataset.
>
> As there has been a recent resurgence of interest in algorithmic developments inspired by a probabilistic approach to meta-learning, we hope that our proposal for exploring the underlying structure of the assumed probabilistic model will give rise to further interesting algorithmic developments. We expect that these developments will make use of inference techniques such as amortized inference, MCMC, classic variational inference (black-box or mean-field), and expectation propagation. For this paper, we resorted to the most tractable approximate inference procedure that is compatible with stochastic gradient descent: maximum-a-posteriori estimation via gradient descent on a loss function taken as the negative log-likelihood (as explained in Section 2).
>
>
> We now discuss the inference procedure of other methods in greater detail.
>
> VERSA [Gor18] makes use of the underlying hierarchical Bayesian model common to recent work in probabilistic meta-learning [Fin17, Gra18, Kim18]. The main difference is the inference procedure, not the model itself. While our approach makes use of traditional statistical inference procedures (i.e., direct optimization of the log-likelihood), VERSA leverages a neural network as a hyper network to learn a mapping from task-specific training data to task-specific parameters. The use of a hyper network to compute the task-specific parameters introduces a trade-off with respect to a purely gradient-based approach: The hyper network introduces additional training overhead as well as the requirement for appropriate architecture design. We do not investigate this trade-off in this work and note that it is most straightforward to adapt MAML-style inference for use in a mixture framework than the hyper-network setup presented in VERSA. Therefore, we do not view it as a detraction of our method that we do not make use of VERSA-style, feedforward computation of task-specific parameters, nor do we view our approach as a direct competitor.

---

### Author Response · Authors · 2018-12-05
**Details on the revised submission**

We sincerely thank the reviewers for feedback on making the submission better. Below, we describe how we have updated the submission to incorporate the suggested revisions.

- To address the concerns of R1 on technical correctness, we have revised the nonparametric mixture section to add more background on the classical inference procedures and traditional trade-offs that motivate our approach. Furthermore, we elaborated on the derivation of our point-estimation procedure to clarify the origin of each objective function in the pseudo-code (Algorithm 3 & Subroutines 4/5).

- We understand that the inclusion of the standard miniImageNet benchmark result in our paper was framed in a manner that does not make our goals transparent, as identified by R2. Accordingly, we have revised our paper to clarify how our results present the state-of-the-art on comparable architectures, and that our focus is on the task-agnostic, continual learning setting.

- We thank R2 for suggesting ablations studies that would improve understanding of the proposed method. Unfortunately, since these specific suggestions were made the morning of the revision deadline, we were unable to include them in the updated version, but will include the following in a future revision:
    - An ablation study for the number of components in the mixture for the standard miniImageNet benchmark.
    - An ablation study for an additional entropy regularizer that encourages cluster differentiation (analogous to the repulsion term in BMAML).
    - Subject to the corresponding authors' release of code, a comparison against BMAML. We would, however, like to point out that R2's claim that the repulsion term will "certainly … do more than having no repulsion" is too strong; this is an empirical question that has not yet been evaluated.
    - An ablation study that better disentangles the effect of model capacity from the mixture approach (by e.g., keeping the number of parameters/filters the same as the mixture). We note that BMAML does not investigate such a baseline, and so the work is subject to R2's criticism: There is a "confounding factor in the increased capacity of the meta-learner".
    - As a baseline such as MAML or BMAML is likely to struggle in the evolutionary setting and suffer from catastrophic forgetting, we will develop a more appropriate baseline to include in this setting. We note that no such method that applies to the task-agnostic continual meta-learning setting with neural networks currently exists. As well, we maintain that the demonstration of a baseline such as MAML failing on an important class of tasks is not "trivial", and we view such a demonstration as an empirical result worthy of note in the meta-learning community.

- We recognize and have fixed, in the updated version of the paper, the mismatch between the experiments section, the figures, and their corresponding captions. (The errata included that an introduction to the standard miniImageNet benchmark was missing, the synthetic experiments description referred to the validation loss as the responsibilities and identified a task change at 2100 instead of 1400, the caption discussed responsibilities not present in the figure, and stated that it reported losses for each cluster but the figure presented the average across clusters.)

- We acknowledge that it was hard to identify the desired differentiation in the previous results for the synthetic regression experiment figure, as it focused on the quantitative improvement of the validation losses. We have thus added a plot for the evolution of cluster assignments for each task over the 3 training phases corresponding to the three underlying task distributions.
- The new figure for regression tasks demonstrates:
    - cluster spawning automatically when the task distribution is changed (in a task-agnostic manner)
    - cluster differentiation across tasks where the 3rd cluster is clearly specialized whereas the 2nd cluster has the highest responsibility for the 2nd task distribution, and so on.

- We moved the evolving miniImageNet experiment figure to the appendix. R2 expressed concerns about mode collapse in this experiment, which we elaborate on in the specific response to R2. We have achieved better differentiation in recent runs of our algorithm by better tuning of the hyperparameters and choice of task-specific image transformation. Nonetheless, we could not satisfactorily assess the effect of these hyperparameters and datasets, and recreate the new figures in time for the revision deadline, and so leave these results to a future version of the paper.

- We have emphasized that our method is the first to address the generalized setting of non-stationary meta-/few-shot learning. In particular, prior work for continual learning does not present suitable datasets or benchmarks for few-shot learning, and current meta-learning work does not avoid catastrophic forgetting.

---

### Meta-Review · Area_Chair1 · 2018-12-16
**Timely idea... but paper lacks in results and conclusive insights.**

**Confidence:** 5
**Recommendation:** Reject

**Metareview:**

This paper is extending the meta-learning MAML method to the mixture case. Specifically, the global parameters of the method are now modeled as a mixture. The authors also derive the elaborate associated inference for this approach.

The paper is well written although Rev2 raises some presentation issues that can surely improve the quality of the paper, if addressed in depth.

The results do not convince any of the three reviewers. Rev3 asks for a clearer exposition of the results to increase convincingness. Rev2 and Rev1 also make similar comments.

Rev1 also questions the motivation of the approach, although the other two reviewers seem to find the approach well motivated. Although it certainly helps to prove the motivation within a very tailored to the method application, the AC weighted the opinion of all reviewers and did not consider the paper to lack in the motivation aspect.

The reviewers were overall not very impressed with this paper and that does not seem to stem from lack of novelty or technical correctness. Instead, it seems that this work is rather inconclusive (or at least it is presented in an inconclusive manner): Rev1 says that the important questions (like trade-offs and other practical issues) are not answered, Rev2 suggests that maybe this paper is trying to address too much, and all three reviewers are not convinced by the experiments and derived insights.

Finally, Rev2 points out some inherent caveats of the method; although they do not seem to be severe enough to undermine the overall quality of the approach, it would be instructive to have them investigated more thoroughly (even if not completely solving them).